# LO-ARMs++: Improving Learning-Order Autoregressive Models for Molecular String Generation

## Abstract

Autoregressive models (ARMs) have become the workhorse for sequence generation tasks, because of their simplicity and ability to exactly evaluate their log-likelihood. Classical Fixed-Order (FO) ARMs factorize high-dimensional data according to a fixed canonical ordering, framing the task as next-token prediction. While a natural ordering exists for text (left-to-right), canonical orderings are less obvious for many data modalities, such as molecular graphs and sequences. Learning-Order (LO) ARMs address this limitation, but their training relies on the optimization of an Evidence Lower Bound (ELBO), rather than on their exact log-likelihood. Therefore, FO-ARMs tend to remain advantageous. In this paper, we introduce LO-ARMs++, an improved version of LO-ARMs, to address this issue through incorporating several technical improvements. We introduce an improved training method called $\alpha$-$\beta$-ELBO, together with network architectural improvements. We demonstrate the general applicability of $\alpha$-$\beta$-ELBO, which yields improvement on the distribution learning metrics on both molecular graph and string generation. Moreover, on the challenging domain of molecular sequence generation, LO-ARMs++ match or surpass state-of-the-art results of Fixed-Order(FO) ARMs on the GuacaMol and MOSES SMILES benchmarks in terms of key metrics for distribution similarity.

Molecular generation in large chemical spaces has important real-world applications such as in drug discovery and material design. While deep generative models for molecular graphs based on diffusion models (Vignac et al., 2023; Eijkelboom et al., 2024; Wang et al., 2025a) are emerging as a promising solution, SMILES (Simplified Molecular Input Line Entry System) string-based methods (Brown et al., 2019; Irwin et al., 2022; Ross et al., 2022; Schwaller et al., 2019) remain popular in practice. This is because SMILES strings are human-interpretable, lead to computationally efficient algorithms compared to handling graph structures, and yield strong performance on key distributional metrics, such as the Fréchet ChemNet Distance (FCD). Technically, SMILES-based models adopt text-based autoregressive architectures (e.g., Recurrent Neural Networks) and inherit their left-to-right generation ordering. However, unlike text data, for which left-to-right appears to be a natural ordering, SMILES data actually encodes tree-like structures and its natural "canonical" ordering between data dimensions is less obvious. Therefore, it is desirable to consider a variant of ARMs that do not treat the ordering as fixed, but rather as a latent random variable that follows a probability distribution that adapts to the evolving state of the generation process.

To address this issue, Wang et al. (2025a) proposed Learning-Order ARMs (LO-ARMs), an ARM variant which can learn human-interpretable autoregressive orderings for image and graph generation and achieves state-of-the-art results on molecular graph generation for distribution similarity and drug-likeness. However, when applied to molecular sequence generation, despite learning human-interpretable orders for molecular sequence generation, LO-ARMs still lag behind Fixed-Order ARMs (FO-ARMs) on FCD.

We provide evidence that that this performance shortfall arises because the order-policy learned with standard LO-ARMs collapses prematurely to a near-deterministic ordering, causing the overall solution to be suboptimal. Indeed, the Evidence Lower Bound (ELBO) optimization, on which LO-ARM training depends, is often complicated by poor local optima and high variance of gradient

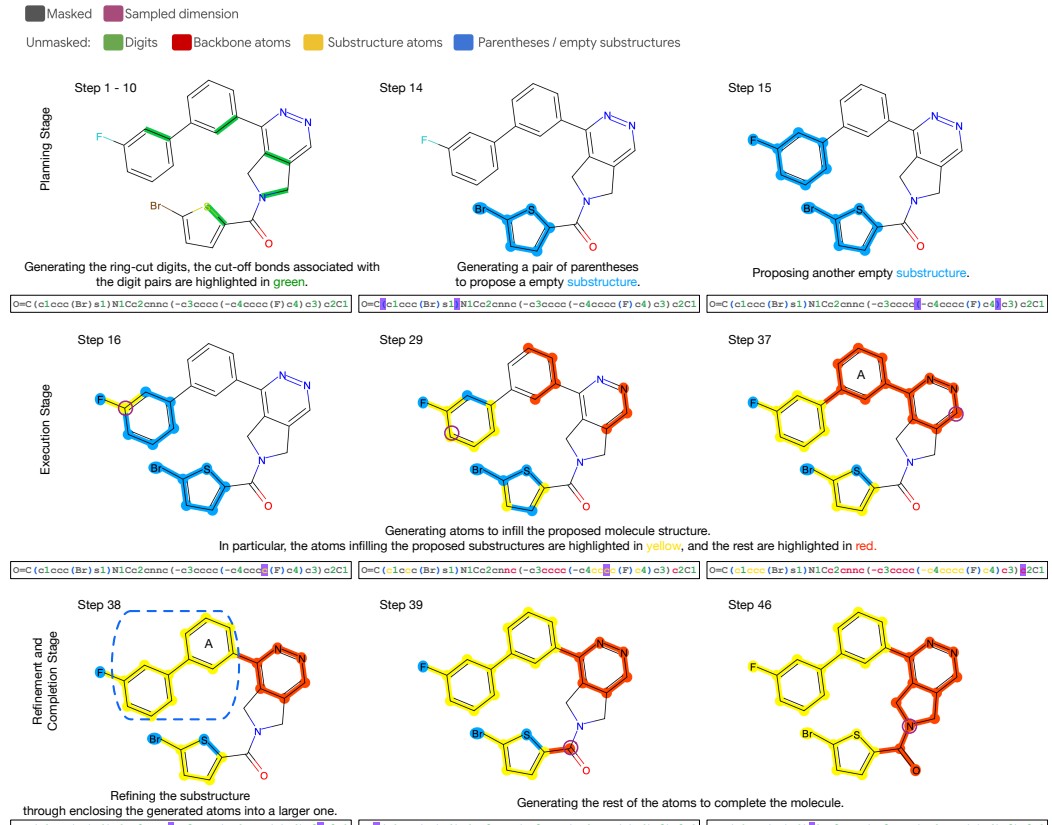

Figure 1: **An example of generating SMILES sample with LO-ARMs++ trained on the GuacaMol dataset**. Our model generates SMILES strings step-by-step, commencing with all dimensions masked (in the figures masked dimensions are colored in grey) and adding one token at a time. First, an *order-policy* selects which dimension to fill, and then a *classifier* determines its value. In this visualization, for each partially generated SMILES string in the subfigures, we highlight the generated components with different colors in the corresponding 2D molecules. The generation proceeds through four phases: 1) Planning (Step 1 to 15): LO-ARM++ first generates pairs of digits (highlighted in green), which represents ring closures. This step determines the number of rings and estimates their potential connections in the molecule. The digits together with their associated ring-cut bonds in the final sample are highlighted in green in the first molecule. Specifically, in this sample, the 5 pairs of digits correspond to exactly 5 rings in the molecule. Then in Step 14 and 15, it proposes two substructures through generating the corresponding pairs of parentheses. In particular, the blues correspond to the dimensions that are enclosed in a proposed substructure but yet to be infilled with atoms. 2) Infilling atoms to the proposed molecule structure (Step 16 to 37). The unmasked atoms are highlighted either in yellow (belonging to a substructure) or red (on the molecule backbone). 3) Refinement (Step 38): In addition to the substructures proposed in Step 14 and 15, LO-ARM++ generates another pair of parentheses to refine the substructures, yielding a larger substructure highlighted in the dotted blue box in Step 38. In particular, as the benzene ring A (labeled in Step 37 and 38) has now been included into the larger substructure, we change its color from red to yellow. 4) Completion (Step 39 to 46): Finally, LO-ARM++ completes the molecule through generating the rest of the atoms on the backbone (highlighted in red). This learned, interpretable ordering is highly consistent: for valid generations containing rings, 94.5% adhere to this overall generation pattern of planning-execution, i.e., generating digits and small pairs of parentheses first followed up generating atoms. Moreover, 80.6% of them contain at least one refinement step at later stages. The full information of generating this sample, including the outputs of the classifier and the order-policy, is provided in Appendix G. Moreover, we provide a sub-optimal ordering learned without the improvements developed in this paper in Appendix H, which generates the pairs of parentheses after all atoms have been generated without any refinement steps.

estimates. The core technical question we address here is whether we can obtain a more efficient order-policy, yielding better generation performance, through improving the training process.

We introduce LO-ARMs++, which resolve the issues encountered in training standard LO-ARMs, in turn yielding better generation performance (see Figure 1). Our main contributions include:

- We introduce $\alpha$-$\beta$-ELBO, an improved training loss, which allows for implementing an exploration-exploitation strategy for unsupervised learning. This forms the basis of the LO-ARMs++ model training procedure, yielding improved generation performance.
- We also propose several network architecture improvements that, when combined with the $\alpha$-$\beta$-ELBO, further stabilize the training of LO-ARMs++.

These improvements can not only yield tighter ELBO on test data, but can also effectively encourage the model to discover more meaningful generation orderings, and consequently achieve better generation performance.

We evaluate our methods on the unconditional generation tasks on the GuacaMol (Brown et al., 2019) the MOSES (Polykovskiy et al., 2020) benchmarks for molecule generation. Our results, measured by distribution learning metrics (e.g., FCD), match or surpass state-of-the-art FO-ARMs relying on a left-to-right generation order. To the best of our knowledge, this is the first discrete diffusion-style model to achieve this level of performance in an important scientific domain.

The paper is organized as follows: Section 1 reviews LO-ARMs. Section 2 details the proposed improvements: identifying issues (Section 2.1), presenting the improved learning loss $\alpha$-$\beta$-ELBO (Section 2.3), and comprehensive measures to improve molecular string generation (Section 3). Section 4 presents the evaluation against the GuacaMol benchmark, including a detailed ablation analysis (Section 4.3).

# 1 BACKGROUND

## 1.1 SMILES-BASED AUTOREGRESSIVE MOLECULE GENERATION

SMILES (Weininger, 1988) is a formal grammar for describing molecule structures with a string of characters. It is generated by performing a depth-first traversal of the molecule's structure and printing the symbols, with parentheses indicating branching points and numbers to denote ring closures. An example of a SMILES string and its corresponding molecule structure are shown in Figure 1.

The SMILES representation allows researchers to directly apply well-developed sequence modeling algorithms to molecule generation. In particular, methods that use ARMs for modeling SMILES strings remain a popular choice (Brown et al., 2019; Schwaller et al., 2019; Irwin et al., 2022; Ross et al., 2022), due to their simplicity and computational efficiency. Despite the rapid progress in molecule generative models, such methods remain state-of-the-art on a number of key metrics such as FCD (Vignac et al., 2023). Specifically, these methods treat SMILES strings as a sequence of characters $\boldsymbol{x} = (x_1, x_2, \ldots, x_L)$ and define a joint probability distribution over $\boldsymbol{x}$: $p_\theta(\boldsymbol{x}) = \prod_{i=1}^{L} p_\theta(x_i|\boldsymbol{x}_{<i})$, where $\boldsymbol{x}_{<i} \triangleq (x_1, \ldots, x_{i-1})$ and $p_\theta(x_i|\boldsymbol{x}_{<i})$ is the conditional distribution with the convention $p_\theta(x_1|\boldsymbol{x}_{<1}) = p_\theta(x_1)$. Typically, these conditional distributions are parameterized with deep learning architectures such as LSTMs and Transformers.

## 1.2 LEARNING-ORDER ARMS

LO-ARMs (Wang et al., 2025a) address a fundamental limitation of ARMs associated with the assumption of a fixed generation order, which may not be efficient for complex data types like graphs and images. LO-ARMs introduce latent variables $\boldsymbol{z} = (z_1, ..., z_L)$ where $z_i$ represents the order index of token $x_i$, i.e., $\boldsymbol{z}$ represents a permutation. They also incorporate a trainable probability distribution that dynamically decides the sampling order of the data dimensions. The log-likelihood of one data point $\boldsymbol{x}$ involves marginalizing over $L!$ permutations, i.e. $\log p_\theta(\boldsymbol{x}) = \log \sum_{\boldsymbol{z}} p_\theta(\boldsymbol{z}, \boldsymbol{x})$, where $p_\theta(\boldsymbol{z}, \boldsymbol{x}) = \prod_{i=1}^{L} p_\theta(z_i|\boldsymbol{z}_{<i}, \boldsymbol{x}_{\boldsymbol{z}_{<i}}) p_\theta(x_{z_i}|\boldsymbol{x}_{\boldsymbol{z}_{<i}})$. Specifically, $p_\theta(z_i|\boldsymbol{z}_{<i}, \boldsymbol{x}_{\boldsymbol{z}_{<i}})$ is called the *order-policy* and $p_\theta(x_{z_i}|\boldsymbol{x}_{\boldsymbol{z}_{<i}})$ is called the *classifier*, and both factors depend on parameters $\theta$ that we want to learn. Since the exact likelihood is intractable (except for very small $L$), the modeling

approach maximizes an ELBO that is obtained by introducing a *variational order-policy* over $\boldsymbol{z}$ that conditions on the full data vector $\boldsymbol{x}$, and has the general form $q_\theta(\boldsymbol{z}|\boldsymbol{x}) = \prod_{i=1}^L q_\theta(z_i|\boldsymbol{z}_{<i}, \boldsymbol{x})$.

## 1.3 TRAINING LO-ARMs WITH VARIATIONAL INFERENCE

To train LO-ARMs, Wang et al. (2025a) established the following ELBO on $\log p_\theta(\boldsymbol{x})$:

$$\log p_\theta(\boldsymbol{x}) \geq \sum_{\boldsymbol{z}} q_\theta(\boldsymbol{z}|\boldsymbol{x}) \log \frac{p_\theta(\boldsymbol{z}, \boldsymbol{x})}{q_\theta(\boldsymbol{z}|\boldsymbol{x})} = \sum_{\boldsymbol{z}} q_\theta(\boldsymbol{z}|\boldsymbol{x}) \sum_{i=1}^L \log \frac{p_\theta(z_i|\boldsymbol{z}_{<i}, \boldsymbol{x}_{\boldsymbol{z}_{<i}}) p_\theta(x_{z_i}|\boldsymbol{x}_{\boldsymbol{z}_{<i}})}{q_\theta(z_i|\boldsymbol{z}_{<i}, \boldsymbol{x})}$$

$$= \sum_{i=1}^L \mathbb{E}_{q_\theta(\boldsymbol{z}_{<i}|\boldsymbol{x})} \left[ \mathbb{E}_{q_\theta(z_i|\boldsymbol{z}_{<i}, \boldsymbol{x})} \left[ \log \frac{p_\theta(z_i|\boldsymbol{z}_{<i}, \boldsymbol{x}_{\boldsymbol{z}_{<i}}) p_\theta(x_{z_i}|\boldsymbol{x}_{\boldsymbol{z}_{<i}})}{q_\theta(z_i|\boldsymbol{z}_{<i}, \boldsymbol{x})} \right] \right] = \sum_{i=1}^L \mathbb{E}_{q_\theta(\boldsymbol{z}_{<i}|\boldsymbol{x})} \left[ F_\theta(\boldsymbol{z}_{<i}, \boldsymbol{x}) \right] \quad (1)$$

and then optimized the ELBO via an unbiased stochastic estimate, which involved sampling one term $i$ uniformly at random in $\{1, ..., L\}$ and its corresponding $\boldsymbol{z}_{<i} \sim q_\theta(\boldsymbol{z}_{<i}|\boldsymbol{x})$ to obtain the negative ELBO unbiased stochastic estimate

$$\mathcal{L}(\theta) = -L F_\theta(\boldsymbol{z}_{<i}, \boldsymbol{x}). \quad (2)$$

Note that, during both training and inference, the generative model $p_\theta$ is conditioned on the sequence length $L$ (i.e., knowing the sequence length before infilling the dimensions). We explain how both standard LO-ARMs and LO-ARMs++ handle variable sequence lengths in Appendix E.1.

## 2 METHODS

Our core research question is whether, in addition to human-interpretability and consistency, LO-ARM can discover "better" order-policies, that in turn yield better generation performance and improved ELBO close to the exact log-likelihood of FO-ARMs. After presenting some issues we have observed when training standard LO-ARMs in Section 2.1, we propose an improved learning loss, $\alpha$-$\beta$-ELBO, mitigating those issues in Section 2.3. In particular, the improvement is inspired by our understanding of LO-ARMs in the setting of Generalized Next-Token-Predictors (NTPs); see Section 2.2. We detail additional improvements to the training algorithm with $\alpha$-$\beta$-ELBO in Section 3. The resulting improvements to LO-ARMs will be denoted as LO-ARMs++.

### 2.1 ISSUES OF LEARNING WITH STANDARD LO-ARMs

When modeling the GuacaMol dataset with the standard LO-ARMs, the variational order-policy $q_\theta(z_i|\boldsymbol{z}_{<i}, \boldsymbol{x})$ converges quickly to a deterministic policy, e.g., in about 100K out of 1.5M training steps, resulting in a greedy order-policy with extreme maximum and minimum logit outputs (as shown in Figure 3). This is because, during training, $q_\theta$ has access to the entire unmasked sequence $\boldsymbol{x}$, yielding faster convergence than the $p_\theta$ network, which is only conditioned on partially observed data $\boldsymbol{x}_{\boldsymbol{z}_{<i}}$. The rapid collapse of the variational order-policy is ultimately harmful, causing several problems: 1) the learned order may converge to a sub-optimal policy (as we can see from the order-policies in Figure 1 and Figure 9), 2) the training may suffer from instability due to excessively large logits in $q_\theta$ (see Appendix D.2).

We therefore aim to design a variational order-policy that maintains a greater degree of randomness for longer, allowing for more robust classifier learning and better exploration over the order-policy. To motivate our solution, we first reformulate LO-ARMs as generalized Next-Token-Predictors (NTPs), which will prove helpful for the subsequent developments.

### 2.2 LO-ARMs ARE GENERALIZED NEXT-TOKEN-PREDICTORS

We rewrite the per-step objective $F_\theta$ defined in Equation (1) as

$$F_\theta(\boldsymbol{z}_{<i}, \boldsymbol{x}) = \mathbb{E}_{q_\theta(z_i|\boldsymbol{z}_{<i}, \boldsymbol{x})} \left[ \log \frac{p_\theta(z_i|\boldsymbol{z}_{<i}, \boldsymbol{x}_{\boldsymbol{z}_{<i}}) p_\theta(x_{z_i}|\boldsymbol{x}_{\boldsymbol{z}_{<i}})}{q_\theta(z_i|\boldsymbol{z}_{<i}, \boldsymbol{x})} \right]$$

$$= \mathbb{E}_{q_\theta(z_i|\boldsymbol{z}_{<i}, \boldsymbol{x})} \left[ \log p_\theta(x_{z_i}|\boldsymbol{x}_{\boldsymbol{z}_{<i}}) \right] - D_{\text{KL}}(q_\theta(z_i|\boldsymbol{z}_{<i}, \boldsymbol{x}) \| p_\theta(z_i|\boldsymbol{z}_{<i}, \boldsymbol{x}_{\boldsymbol{z}_{<i}})). \quad (3)$$

The first term corresponds to the cross-entropy loss optimizing the classifier. Specifically, in the LO-ARM case, $q_\theta(z_i|\boldsymbol{z}_{<i}, \boldsymbol{x})$ samples the next dimension to generate, and the classifier $\log p_\theta(x_{z_i}|\boldsymbol{x}_{<i})$ predicts the value. From this perspective, $q_\theta(z_i|\boldsymbol{z}_{<i}, \boldsymbol{x})$ effectively reweights the cross-entropy losses across the remaining dimensions. Equivalently, we can interpret $q_\theta$ as a *problem setter* for the classifier, selecting which dimension the classifier must predict next.

The above view unifies FO- and AO-ARMs: 1) in AO-ARMs, $p(z_i|\boldsymbol{x}_{\boldsymbol{z}_{<i}}) = q(z_i|\boldsymbol{z}_{<i}, \boldsymbol{x}) = q(z_i) = \text{Uniform}(\{1 \dots L\} \setminus \boldsymbol{z}_{<i})$, and the classifier must be as general as possible, since it faces a uniform distribution over the remaining dimensions on which it will be required to make a prediction. By contrast 2) in FO-ARMs, $q(z_i) = \delta(z_i = k), k \in \{1 \dots L\} \setminus \boldsymbol{z}_{<i}$, and the classifier needs only to predict a single known dimension at each step. For left-to-right ARMs, $k = i$. Note that, in both cases, the KL terms zero out, and only the cross-entropy terms are left. LO-ARMs generalize FO- and AO-ARMs by using learnable and context-dependent distributions $q(z_i|\boldsymbol{z}_{<i}, \boldsymbol{x})$ and $p(z_i|\boldsymbol{x}_{\boldsymbol{z}_{<i}})$.

## 2.3 $\alpha$-$\beta$-ELBO

From the perspective of variational inference (i.e., Equation (3)), the FO-ARM can yield the exact log-likelihood, because 1) its KL divergence is always zero, and 2) the variance induced by the degenerate order policy (i.e. $\delta(z_i = k)$) in the cross-entropy term is also zero. In contrast, while the KL term in AO-ARMs is also zero, they maximize the cross-entropy variance by sampling uniformly over all remaining dimensions.

Inspired by these insights, we motivate our improvements to achieve a tighter ELBO from two high-level requirements: 1) to efficiently minimize the KL divergence between $p_\theta(z_i|\boldsymbol{x}_{\boldsymbol{z}_{<i}})$ and $q_\theta(z_i|\boldsymbol{z}_{<i}, \boldsymbol{x})$, and 2) to reduce the variance of gradient estimates incurred by sampling $q_\theta(z_i|\boldsymbol{z}_{<i}, \boldsymbol{x})$. These yield the following modified objective function with respect to the generalized NTP $F_\theta$, which we call $\alpha$-$\beta$-ELBO:

$$F_\theta = \underbrace{\mathbb{E}_{q_\theta(z_i|\boldsymbol{z}_{<i}, \boldsymbol{x})}\left[\log p_\theta(x_{z_i}|\boldsymbol{x}_{\boldsymbol{z}_{<i}})\right]}_{(a)} - \underbrace{\beta D_{\text{KL}}(q_\theta(z_i|\boldsymbol{z}_{<i}, \boldsymbol{x}) \| p_\theta(z_i|\boldsymbol{z}_{<i}, \boldsymbol{x}_{\boldsymbol{z}_{<i}}))}_{(b)} + \underbrace{\alpha H\left[q_\theta(z_i|\boldsymbol{z}_{<i}, \boldsymbol{x})\right]}_{(c)}$$

(4)

$$= \mathbb{E}_{q_\theta(z_i|\boldsymbol{z}_{<i}, \boldsymbol{x})}\left[\log p_\theta(x_{z_i}|\boldsymbol{x}_{<i})\right] + \beta\mathbb{E}_{q_\theta(z_i|\boldsymbol{z}_{<i}, \boldsymbol{x})}\log p_\theta(z_i|\boldsymbol{z}_{<i}, \boldsymbol{x}_{\boldsymbol{z}_{<i}}) + (\alpha + \beta)H\left[q_\theta\right], \quad (5)$$

where $\beta \geq 1$ and $\alpha \geq 0$, and $H\left[q_\theta(z_i|\boldsymbol{z}_{<i}, \boldsymbol{x})\right] = H[q_\theta] = -\mathbb{E}_{q_\theta}[\log q_\theta]$ is the entropy of $q_\theta$.

We now show how these components address the issues observed in Section 2.1. First, component (c) implements the standard maximum entropy regularization on $q_\theta$. Second, setting $\beta \geq 1$ in (b) upweights the KL distillation from $q_\theta$ to $p_\theta(z_i|\boldsymbol{x}_{\boldsymbol{z}_{<i}})$. Moreover, as the KL term already implicitly imposes an entropy regularization on $q_\theta$, the total entropy regularization imposed on $q_\theta$ is controlled by $\alpha + \beta$, see Equation (5). This entropy term is crucial during early stages of training, since it causes the variational order-policy to maintain high entropy when $\alpha + \beta$ is large, preventing premature collapse and presenting a diversity of prediction problems to the classifier. Additionally, the KL term encourages the model order-policy $p_\theta(z_i|\boldsymbol{x}_{\boldsymbol{z}_{<i}}, \boldsymbol{z}_{<i})$ to imitate the variational order-policy $q_\theta$. These dual goals mirror the use of maximum entropy policies in reinforcement learning to balance exploration and exploitation (Mnih et al., 2016; Haarnoja et al., 2017).

Note that, while components (a) and (b) together resemble a $\beta$-VAE (Higgins et al., 2017), a key difference here is that we are working with discrete distributions, which may not always cover the full support of data dimensions, resulting in collapsed, deterministic policies. Therefore, we argue that the maximum entropy regularization on $q_\theta$ is essential. We provide additional ablation for this argument in Section 4.3.

### 2.3.1 EXPLORATION-EXPLOITATION THROUGH ANNEALING $\alpha$ AND $\beta$

The $\alpha$-$\beta$-ELBO generalizes the standard ELBO defined in Equation (3) in the following ways: 1) when $\alpha = 0, \beta = 1$, $\alpha$-$\beta$-ELBO recovers the standard ELBO; 2) $\alpha > 0, \beta = 1$ corresponds to training with standard maximum entropy regularization on $q_\theta$.

We implement an exploration-exploitation optimization strategy, inspired by reinforcement learning, through applying two annealing schedules to $\alpha$ and $\beta$ respectively, decaying an initial $\alpha > 0$ down to

0 and an initial $\beta > 1$ down to 1. In the exploration stage, where $\alpha > 0$ and $\beta > 1$, we want to present the classifier with a diversity of learning problems and explore over model order-policy with a high entropy variational distribution $q_\theta$, while ensuring that $p_\theta(z_i | \boldsymbol{x}_{<i}, \boldsymbol{z}_{<i})$ tracks $q_\theta$. Since our ultimate objective is to optimize the ELBO, in the exploitation stage we shift $\alpha$-$\beta$-ELBO back to the standard ELBO with $\alpha = 0$ and $\beta = 1$. During this latter phase, we further optimize the reweighted cross entropy term (i.e., (a) in Equation (4)) with the more stable $q_\theta$. We detail the annealing schedules in Appendix E.3. Moreover, we demonstrate the general applicability of $\alpha$-$\beta$-ELBO through applying it to both molecular graphs (i.e., GuacaMol and ZINC250k graphs) and strings (i.e., GuacaMol and MOSES SMILES) generation, and on both domains, $\alpha$-$\beta$-EBLO improves generation performance (see Appendix C).

## 3    LO-ARMs++ FOR MOLECULAR STRING GENERATION

We implement our LO-ARMs with Transformer to fully utilize modern hardware accelerators, and directly compete against FO-ARMs on sequence generation, specifically SMILES strings, implemented with two main architectures, i.e., Transformer and Recurrent Neural Networks. In this section, we introduce several changes complementary to $\alpha$-$\beta$-EBLO, which facilitate training Transformer-based ARMs. We first introduce a novel preprocessing scheme in Section 3.1. Then we describe how to deal with strings of variable length in Transformer in Section 3.2.

We leverage the network architecture introduced in (Wang et al., 2025a). Specifically, we collocate the classifier $p_\theta(x_{z_i} | \boldsymbol{x}_{<i}, \boldsymbol{z}_{<i})$ and the model order-policy $p_\theta(z_i | \boldsymbol{x}_{<i}, \boldsymbol{z}_{<i})$ through a shared backbone, and use a separate neural network to implement $q_\theta(z_i | \boldsymbol{z}_{<i}, \boldsymbol{x})$. Both networks are implemented with a transformer (Vaswani et al., 2017). In particular, the model network consists of 18 attention layers, while the $q_\theta$ network remains quite lightweight, consisting of 6 attention layers. We detail the network architectures in Appendix E. Moreover, the training algorithm remains largely the same as in Wang et al. (2025a) besides the changes introduced in the Method section Section 2.

### 3.1    PREPROCESSING SMILES STRINGS WITH PREFIX TOKENIZATION

We employ the prefix tokenization (Wang et al., 2025b) to preprocess the SMILES strings. Specifically, instead of parsing individual parentheses as tokens, the prefix tokenization represent matching parenthesis pairs as individual tokens. These pairs are formatted as @N, where $N$ is the size of the parenthesis pair (the number of tokens between the matching parentheses, including the right parenthesis). An example of preprocessed data is provided in Appendix A.1. Note that, compared to other tokenization methods (e.g., SELFIES (Krenn et al., 2020)), this prefix tokenization introduces minimal changes to the raw SMILES strings. In fact, our ablation analysis (Appendix D.3) shows that, compared to the standard character-based tokenization, this prefix tokenization does not affect FO-ARM's generation performance on distributional metrics (i.e., FCD). Note that, the character-based tokenization implicitly assumes a left-to-right token order through parsing individual parentheses as tokens, and therefore, it is compatible with fixed left-to-right ARMs. From this perspective, this prefix tokenizer is best viewed as a compatible one to LO-ARMs or other diffusion models, which do not assume a fixed left-to-right order while preserving the original data distribution.

### 3.2    STABLE GENERALIZATION FOR MODELING SEQUENCES OF VARIABLE LENGTHS

A subtle problem we encountered during development was that the standard attention dropout employed in LO-ARM transformers is disruptive to training (see Appendix D), i.e., directly applying dropout to attention scores $\text{Attention}(Q, K, V) = \text{Dropout}\left(\text{softmax}\left(\frac{QK^T}{\sqrt{d_k}}\right)\right) \cdot V$, where $Q, K, V$ are the queries, keys and values respectively. We hypothesize that, because LO-ARMs model molecular strings of variable lengths and the padding dimensions are zeroed out in the attention score matrix, if we directly dropout the attention scores, the model will confuse with the dropped out dimensions and the padding dimensions, which are both zeros. We fix this issue by applying dropout on the output of the outer multiplication of the value matrix and the corresponding attention scores, i.e., $\text{Attention}(Q, K, V) = \text{Dropout}(\text{softmax}\left(\frac{QK^T}{\sqrt{d_k}}\right) \cdot V)$. This simple yet effective fix yields stable generalization during training and improved generation performance at test time (see the ablation analysis in Section 4.3).

Moreover, we find that applying the improved dropout to the model network (i.e., $p_\theta$) also encourages the variational distribution $q_\theta$ to be more uniform (see Figure 2). Therefore, to simplify the configuration of hyperparameters, we choose to regularize the $q_\theta$ network only via the global KL and maximum entropy regularization, and apply extra regularization on the $p_\theta$ network with the improved dropout.

## 4 RESULTS AND ANALYSIS

### 4.1 EXPERIMENT SETUP

We evaluate our methods on the unconditional molecule generation tasks for the GuacaMol (Brown et al., 2019) and the MOSES (Polykovskiy et al., 2020) benchmarks. These are standard benchmarks for evaluating generative models for molecule generation, with a particular focus on distribution learning (Irwin et al., 2022; QIN et al., 2025; Schwaller et al., 2019). For each benchmark, we use the standard dataset preprocessing and splits for training as well as the standard evaluation setup, including metrics and evaluation tools provided in the literature. We selected these two benchmarks because: 1) SMILES strings encode graph-structured molecules as flat sequences, meaning a natural token generation order is less obvious than in natural language 2) Autoregressive models (ARMs) with a left-to-right sequence are a robust baseline for SMILES synthesis, consistently outperforming other methods on distributional metrics. This implies they also yield a strong log-likelihood evaluation, which acts as a clear target for us to improve LO-ARMs with $\alpha$-$\beta$-ELBO. 3) These two benchmarks together construct a comprehensive evaluation matrix for LO-ARMs++. Specifically, they cover both canonicalized (i.e., GuacaMol) and non-canonicalized (i.e., MOSES) SMILES strings, and provide good variability in molecule complexity (Bagal et al., 2022) (i.e., MOSES molecules generally have shorter average SMILES lengths and less dispersed property distributions than GuacaMol). 4) Practically, we also hope to demonstrate the usefulness of LO-ARMs++ through enriching the toolkit for real-world applications (e.g., drug discovery).

We evaluate them on two key aspects: 1) For individual molecules, we assess their chemical validity, uniqueness and novelty. 2) Distributional similarity between generated and ground truth samples (e.g., Fréchet ChemNet Distance (FCD)). Specifically, for each benchmark, we directly employ its standard evaluation metrics on distribution learning, detailed in Appendix A.1. To conduct our evaluation, for each of our own baselines, we sample 5 individual batches of generated samples, each of which contains $16,384$ molecules for the GuacaMol benchmark or $32,768$ molecules for the MOSES benchmark, and we report both the mean and standard deviation for each model. For other baselines in the literature (e.g., VAE and LSTM (Brown et al., 2019)), which do not report standard deviations, we directly cite their reported results.

### 4.2 MAIN RESULTS ON DISTRIBUTION LEARNING

To evaluate the order policy, we add two baselines to the LSTM-ARMs: 1) a Transformer FO-ARM, to match our Transformer-based LO-ARMs, and 2) AO-ARM (Any-Order), where both the variational ($q_\theta$) and model ($p_\theta$) order policies are uniform. Additionally, for both GuacaMol and MOSES SMILES benchmarks, we report the results of VAE and LSTM/RNN, as recent literature (e.g., (QIN et al., 2025; Vignac et al., 2023)) recognizes them as top-performing models for these benchmarks. In Table 1 and Table 2, we restrict our comparison to SMILES-based methods, as graph-based models (e.g., (QIN et al., 2025; Vignac et al., 2023)) still lag behind SMILES-based ones by a large margin. To better situate LO-ARMs++ in the literature, we compare it against graph-based methods in Table 4 and Table 5. Finally, to highlight the effectiveness of the $\alpha$-$\beta$-ELBO, we apply prefix-tokenization and the dropout patch to all our models, including AO-ARMs, Transformer FO-ARMs, and LO-ARMs, unless specified. A full ablation analysis of the individual techniques introduced in Section 3 is provided in Section 4.3 and Appendix D.3.

**Results on the GuacaMol benchmark.** In Table 1, we observe that LO-ARM++ significantly outperforms the standard LO-ARM in terms of FCD and KL divergence. This substantial improvement demonstrates that our enhancements effectively tighten the ELBO. Furthermore, LO-ARM++ outperforms LSTM, achieving state-of-the-art results on both distributional metrics.

Table 1: Molecule generation on GuacaMol SMILES dataset. We directly report other methods results on the following metrics: **V**alidity, **U**niqueness, **N**ovelty, FCD and KL divergence). **V.N.** means both valid and unique, and **V.U.N.** means samples are valid, unique and novel. The metrics are calculated on samples generated by each method. The random sampler uniformly samples the test set. Bold and underlined numbers indicate the best and second-best results, respectively. An extended result table is provided in Table 4.

| Method | Tokenization | V.%↑ | V.U.%↑ | V.U.N.%↑ | FCD↑ | KL Div.↑ |
|---|---|---|---|---|---|---|
| Random sampler | - | 100.0 | 99.7 | 0.0 | 92.9 | 99.8 |
| AAE | Standard | 82.2 | 82.2 | 88.0 | 52.9 | 88.6 |
| VAE | Standard | 87.0 | 86.9 | 84.7 | 86.3 | 98.2 |
| LSTM | Standard | 95.9 | 95.9 | 87.5 | 91.3 | 99.1 |
| Our Results | | | | | | |
| AO-ARM | Prefix | $63.3 \pm 0.3$ | $63.2 \pm 0.3$ | $62.8 \pm 0.2$ | $72.1 \pm 0.7$ | $91.7 \pm 0.5$ |
| FO-ARM | Standard | $\mathbf{98.1 \pm 0.2}$ | $\mathbf{98.0 \pm 0.3}$ | $\mathbf{88.6 \pm 0.3}$ | $87.0 \pm 0.4$ | $99.1 \pm 0.1$ |
| FO-ARM | Prefix | $83.3 \pm 0.7$ | $83.1 \pm 0.2$ | $82.8 \pm 0.3$ | $87.2 \pm 0.2$ | $99.1 \pm 0.2$ |
| LO-ARM | Standard | $94.2 \pm 0.1$ | $94.0 \pm 0.1$ | $90.2 \pm 0.4$ | $36.6 \pm 0.2$ | $40.0 \pm 1.1$ |
| LO-ARM | Prefix | $92.6 \pm 0.3$ | $92.6 \pm 0.3$ | $87.1 \pm 0.3$ | $79.4 \pm 0.3$ | $98.3 \pm 0.2$ |
| LO-ARM++ | Prefix | $93.9 \pm 0.2$ | $93.9 \pm 0.2$ | $85.9 \pm 0.3$ | $\mathbf{91.4 \pm 0.1}$ | $\mathbf{99.2 \pm 0.1}$ |

Next, data in Table 1 reveals that both FO-ARMs (either LSTM or Transformer) and LO-ARMs outperform AO-ARM on FCD, emphasizing that an ordering strategy is crucial for generating SMILES sequences. Furthermore, LO-ARM++ outperforms the Transformer FO-ARM in uniqueness, novelty, FCD and KL divergence. This suggests that, with the same architecture, learning a data-dependent generation order from data is more sample efficient than using a fixed one.

Thirdly, LO-ARM++ learns a consistent, human-interpretable generation order without specific inductive biases (Figure 1). The typical learned process is: 1) Estimate the molecular structure (rings and connections) by first generating digit tokens for ring enclosures and cuts and proposing substructures via pairs of parentheses. 2) Infill the structure, prioritizing non-aromatic tokens over aromatic ones. 3) Refine substructures (Step 38 in Figure 1) by enclosing initial proposals from Stage 1 into larger ones. 4) Complete the molecule by infilling the remaining atom dimensions. The interpretability of these learned orderings allows us to verify patterns with simple rules (Appendix D.1). This interpretable ordering shows high consistency: for valid generations containing rings, $94.5\%$ follow this structure-first pattern, and $80.6\%$ of these refine the substructures at least once.

The generation order of LO-ARM++ notably differs from the standard LO-ARMs without $\alpha$-$\beta$-ELBO (Figure 9) in two ways: 1) The improved order-policy proposes substructures at the beginning of the generation process, rather than finalizing them last. 2) It is also able to refine substructures later in the generation. This suggests the improved order policy generalizes better, as it is more dynamic and can utilize local context more efficiently, meeting the primary goals of our development.

**Greediness of the learned order policy.** Finally, as see in Figure 2, training with $\alpha$-$\beta$-ELBO loss makes the variational order-policy $q_\theta$ less greedy (i.e., it has larger entropy). We now show that this property transfers to the model order-policy $p_\theta(z_i | \boldsymbol{x}_{\boldsymbol{z}_{<i}}, \boldsymbol{z}_{<i})$, yielding a less greedy order-policy for generating new samples. To do this, for each sample's generation trajectory, we calculated per-step correlation coefficients between the order policy probabilities and the classifier entropy (our certainty measure) over all masked dimensions. We then performed one-sample t-tests on each sequence to obtain a mean and a significance level. A higher negative mean correlation between the two quantities means the order policy is greedier, as it prioritizes dimensions with higher certainty (i.e., lower classifier entropy). For samples generated with LO-ARM++, we found that only $49.2\%$ ($p < 0.05$) exhibited a negative mean correlation, compared with $73.1\%$ ($p < 0.05$) reported for standard LO-ARMs in Wang et al. (2025b). This confirms that the order-policy learned with LO-ARM++ has a less greedy generation strategy than standard LO-ARMs.

**Results on the MOSES benchmark.** LO-ARM without the $\alpha$-$\beta$-ELBO exhibits significantly inferior performance compared to LO-ARM++ on distributional metrics (e.g., FCD 4.39 vs 0.14). This

Table 2: Molecule generation on MOSES SMILES dataset. In addition to Validity, Novelty and FCD, we report the results on the standard distributional metrics introduced in MOSES. E.g,, SNN (nearest neighbor similarity), IntDiv (Internal diversity), and Frag (fragment similarity). Note that, unlike GuacaMol's FCDs, the standard MOSES's FCDs are unnormalized (smaller is better). Moreover, metrics of lipophilicity (logP), Synthetic Accessibility (SA), and Quantitative Estimate of Druglikeness (QED) measure the Wasserstein-1 distance between generated and test set molecule distributions. "-" means not applicable due to lack of reports. A full explanation of these metrics are detailed in Appendix A.1 An extended result table is provided in Table 5.

| Method | Validity↑ | Novelty↑ | FCD↓ | SNN ↑ | Frag↑ | Scaf ↑ | IntDiv2↑ | logP↓ | SA↓ | QED↓ |
|---|---|---|---|---|---|---|---|---|---|---|
| Random sampler | 1.0 | 0.0 | 0.01 | 0.6419 | 1.0 | 0.991 | 0.851 | 0.0 | 0.0 | 0.0 |
| VAE | **0.977** | 0.695 | 0.10 | **0.626** | 0.999 | **0.939** | 0.850 | **0.023** | 0.014 | **0.0006** |
| | ± 0.001 | ± 0.007 | ± 0.01 | ± 0.001 | ± 0.000 | ± 0.002 | ± 0.000 | - | - | - |
| CharRNN | 0.975 | 0.842 | **0.07** | 0.602 | **1.0** | 0.924 | 0.850 | 0.057 | 0.016 | 0.0022 |
| | ± 0.026 | ± 0.051 | ± 0.02 | ± 0.021 | ± 0.0 | ± 0.006 | ± 0.001 | - | - | - |
| Our Results | | | | | | | | | | |
| LO-ARM | 0.663 | **0.976** | 4.39 | 0.567 | 0.954 | 0.924 | **0.857** | 0.645 | 0.18 | 0.078 |
| | ± 0.037 | ± 0.013 | ± 0.13 | ± 0.010 | ± 0.021 | ± 0.003 | ± 0.011 | ± 0.006 | ± 0.010 | ±0.0006 |
| LO-ARM++ | 0.946 | 0.801 | 0.14 | 0.611 | **1.0** | 0.929 | 0.854 | 0.047 | **0.012** | 0.0015 |
| | ± 0.014 | ± 0.021 | ± 0.01 | ± 0.000 | ± 0.0 | ± 0.003 | ± 0.001 | ± 0.002 | ± 0.001 | ±0.0003 |

disparity is likely attributable to the non-canonicalized nature of the MOSES SMILES strings, where the increased randomness in ordering leads to higher variance in learning the generation orderings. The ability of LO-ARM++ to maintain satisfactory performance on these metrics, demonstrates the efficacy of the $\alpha$-$\beta$-ELBO. Second, LO-ARM++ performs comparably to state-of-the-art models like VAE and CharRNN/LSTM on distribution learning metrics. This is because the MOSES benchmark is relatively simple, with shorter sequences and less dispersed property distributions, making sophisticated expressivity less critical. However, when applied to the GuacaMol benchmark, LO-ARM++ achieves state-of-the-art performance. Finally, LO-ARM++ outperforms CharRNN on seven of eight distributional metrics, with only a small FCD deficit. This confirms that learning generative orderings improves sample efficiency in distribution learning, even with non-canonicalized SMILES strings.

## 4.3 ABLATION ANALYSIS

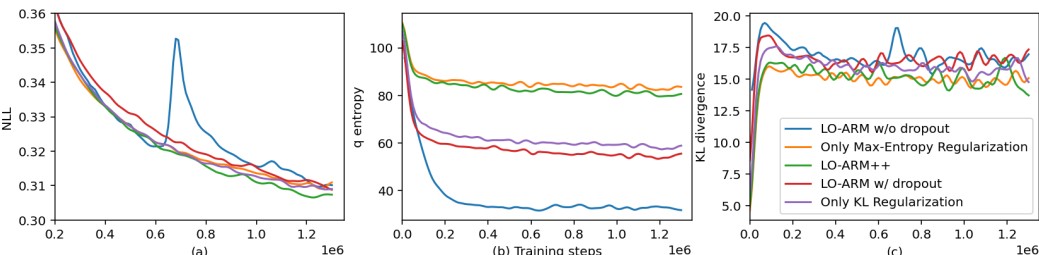

(a)  (b) Training steps  (c)

Figure 2: **Ablation analysis on the effectiveness of KL and maximum entropy regularizations and the improved dropout**. All the metrics evaluated against the test set.

We ablate each improvement by analyzing the following cases: 1) LO-ARM with improved attention dropout, 2) LO-ARM without improved attention dropout, 3) LO-ARM++ with only maximum entropy regularization ($\alpha = 0.075, \beta = 1$), 4) LO-ARM++ with only KL regularization ($\alpha = 0, \beta = 1.075$), and 5) LO-ARM++ with full regularization ($\alpha = 0.025, \beta = 1.05$). Cases 3), 4) and 5) all use improved attention dropout. Additionally, we control the total entropy penalization and vary the KL regularization weight (as shown in Equation (5)). To isolate each component's contribution, we set $\alpha$ and $\beta$ constant without annealing during training in this ablation analysis. First, Figure 2(b) shows that applying attention dropout to the $p_\theta$ network regularizes $q_\theta$. Without improvements, the standard LO-ARM's variational order-policy $q_\theta$ converges to being deterministic very quickly with the lowest entropy. The entropy of $q_\theta$ also increases with a larger $\beta$, confirming the effectiveness of maximum entropy regularization. Next, (c) shows that KL regularization encouraged lower KL divergence between $q_\theta(z_i|z_{<i}, \boldsymbol{x})$ and $p_\theta(z_i|\boldsymbol{x}_{z_{<i}})$, suggesting the model order-policy can imitate $q_\theta$

well. However, KL regularization alone ($\alpha = 0, \beta = 1.075$) does not yield the lowest KL divergence; instead, a combination of both regularization terms (LO-ARM++ with $\alpha = 0.025, \beta = 1.05$) does. This is likely because the effective maximum entropy regularization in LO-ARM++ makes the policy easier for the model to track. Finally, combining all improvements, LO-ARM++ yields the best negative log-likelihood (NLL). We observe the standard LO-ARM's NLL is unstable, spiking at 700k steps. This instability is likely because a deterministic $q_\theta$ yields extreme logit outputs. To confirm this, we visualize the evolution of the maximum and minimum $q_\theta$ logits during training in Appendix D.2.

## 5 RELATED WORK

**Learning Non-Monotonic Autoregressive Orderings** has been studied extensively in recent literature (e.g., Li et al., 2021; Gu et al., 2019; Welleck et al., 2019), and is challenged by the need to find an optimal permutation from a factorial ($L!$) search space, where $L$ is the sequence length. Some methods reduce this space with domain-specific assumptions (Welleck et al., 2019; Gu et al., 2019). Specifically, Welleck et al. (2019) proposes a tree-based recursive generation method to learn arbitrary generation orders, and Gu et al. (2019) combines 1) pretraining with prescribed base orderings and 2) fine-tuning those orderings with Searched Adaptive Order (SAO). Moreover, both Variational Order Inference (VOI) (Li et al., 2021) and LO-ARMs (Wang et al., 2025a) learns orderings with a variational policy. The main difference is that SAO uses a policy gradient procedure and requires optimizing a complex variational ordering distribution that has an intractable normalizing constant and requires a Bethe-type approximation. In contrast, the variational distribution in LO-ARMs (Wang et al., 2025a) and LO-ARMs is fully tractable, allowing for fast, exact, and unbiased gradient-based optimization of the ELBO using REINFORCE leave-one-out.

**Discrete Diffusion and Its Application to Molecular Graph Generation.** Discrete diffusion models (Vignac et al., 2023; QIN et al., 2025) have become a popular alternative to molecular graph generation. LO-ARMs++ also relates to discrete diffusion models based on absorbing or masked diffusion (Austin et al., 2021; Lou et al., 2024; Shi et al., 2024; Sahoo et al., 2024; Ou et al., 2024). Similar to masked diffusion, our discrete architecture treats ungenerated dimensions as masked. The key difference is that we learn a non-uniform, data-dependent generation order via a neural order-policy. Masked diffusion and AO-ARMs (Hoogeboom et al., 2022), in contrast, use a completely random order. Additionally, our approach defines only a backward generative model to sample from a fully masked state, learning a variational order distribution ($q_\theta$) from the data instead of specifying a forward noising process. There are also works on masked diffusions that consider adaptive inference or sampling strategies for unmasking dimensions, such as based on top probability (Zheng et al., 2024) and top probability margin (Kim et al., 2025). Our approach differs since it trains from data a strategy that unmasks the dimensions one at a time.

## 6 CONCLUSION

We have introduced LO-ARMs++, an improved version of LO-ARMs, which allows for learning more data efficient generation orderings in distribution learning. Evaluated on the GuacaMol dataset, with the improved techniques, LO-ARMs++ match or surpass the standard ARMs with fixed generation order. Furthermore, we showed that LO-ARMs++ can still learn human-interpretable and consistent context-dependent generation orders. We found that LO-ARMs++ are particularly useful for data without obvious canonical generation orders, and we will further investigate its practical usefulness in modeling more complex data, e.g., protein sequences. Finally, since LO-ARMs are generalized next-token-predictors it would be interesting to theoretically investigate whether they can be more robust, than fixed order ARMs, to existing criticisms associated with modeling human thought (Bachmann & Nagarajan, 2024).

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

## A  PREFIX TOKENIZATION AND EVALUATION METRICS

### A.1  PREPROCESSING WITH PREFIX TOKENIZATION AND DATASET SUMMARY

We preprocess SMILES strings in two main steps. First, we apply standard tokenization using a widely adopted regular expression (Irwin et al., 2022; Schwaller et al., 2019). Second, to address the strict paired-parenthesis constraint in SMILES grammar— a challenge for models without fixed left-to-right ordering (like LO-ARM or diffusion-based methods) which contrasts with simpler handling in autoregressive generation—we represent parenthesis pairs as individual tokens. Specifically, these pairs are formatted as @N, where $N$ is the size of the matching pairs (the number of tokens between the brackets, including the right parenthesis). Using these new tokens, we then transform the raw SMILES strings into a prefix notation, where each @N parenthesis token precedes the substructure or branch it encloses. An example of this transformation is provided below. It is important to note that this prefix transformation for parentheses is bijective and lossless, and therefore, we can fully recover the original SMILES strings from their corresponding prefix notations.We provide an ablation analysis on different tokenization algorithms in Appendix D.3. Following this transformation, we filter out low-frequency tokens (fewer than 100 occurrences) and the corresponding samples containing them. The preprocessed dataset is summarized in Table 3.

Raw SMILES: `CCOc1ccc(S(=O)(=O)Nc2ccccc2Cl)cc1`
Converted: `CCOc1ccc@20S@3=O@3=ONc2ccccc2Clcc1`

Following this transformation, we filter out low-frequency tokens (fewer than 100 occurrences) and the corresponding samples containing them. The preprocessed dataset is summarized in Table 3. After filtering, the vocabulary size is almost halved while the dataset remains the same scale, only fewer than 1000 samples were filtered out. Note that, although the training set is filtered, we always use the unfiltered ground truth data as the reference distribution when evaluating on the distributional metrics.

Table 3: Dataset statistics before and after filtering. Both cases use the augmented vocabulary and transform SMILES strings with prefix notation described in Section 3.1.

|              | #training samples | #validation samples | #test samples | Vocabulary size |
|--------------|-------------------|---------------------|---------------|-----------------|
| Raw dataset  | 1273114           | 79568               | 238706        | 203             |
| Preprocessed | 1272277           | 79506               | 238538        | 129             |

### A.2  SUMMARY OF DISTRIBUTIONAL METRICS

In this paper, we focus on the generation performance of our models in terms of distribution learning, and therefore, we mainly compare to other models on the distributional metrics employed in the literature of molecule generation. We note that the GuacaMol and MOSES benchmarks use two different set of distributional metrics, and we detail the corresponding metrics in this section.

### A.3  THE GUACAMOL METRICS

- Preuer et al. (2018) introduced Fréchet ChemNet Distance (FCD) as a measure of how close distributions of generated samples are to the distribution of molecules in a reference set. The FCD is determined from the hidden representation of molecules in a neural network

called ChemNet trained for capturing important chemical and biological features, similarly to the Fréchet Inception Distance (FID) in image generation. Note that, FCD is sample-size-dependent, and for all FCD evaluations against the GuacaMol benchmark, the standard in the literature is only using 10000 samples for both the generated and ground truth samples. Moreover, usually better generation performance yields smaller FCD, but the GuacaMol benchmark normalizes FCD, given by $S = \exp(-0.2 \cdot \text{FCD})$.

- KL divergence. For this task, a set of physicochemical descriptors calculated with the RDKIT for both the sampled and the reference set, and then the distributions of these descriptors is computed via kernel density esitmation for continous discriptors, or as a histogram for discrete desriptors. Finally, the KL divergence $D_{\mathcal{D}_{\text{test}}KL,i}$ of each descriptor $i$ is aggregated through $S = \frac{1}{k}\sum_i^k \exp(-D_{\text{KL},i})$.

### A.4 THE MOSES METRICS

The MOSES benchmark also employs FCD as one of the main distributional metric. Two key differences from the use of FCD in GuacaMol 1) The MOSES FCDs are unnormalized raw values, and therefore, the smaller the better. 2) MOSES suggests using 30000 samples in both the generated and the reference sets, a larger sample size than that in GuacaMol.

In addition to FCD, here is a list of additional chemical specific metrics employed in the benchmark, including

- Fragment similarity (Frag) and Scaffold similarity (Scarf), which are cosine distances between vectors of fragments fragment or scaffold frequencies correspondingly of the generated and test sets.
- Nearest neighbor similarity (SNN) is the average similarity of generated molecules to the nearest molecule from the test set.
- Internal diversity (IntDiv) is an average pairwise similarity of generated molecules.
- Additionally, for comparison of molecular properties, we also compute the Wasserstein-1 distances between distributions of molecules in the generated and test sets, for lipophilicity (logP), Synthetic Accessibility (SA), Quantitative Estimation of Drug-likeness (QED).

## B EXTENDED RESULTS ON SMILES STRING GENERATION

We provide the extended tables for the evaluation against GuacaMol ( Table 4) and that against MOSES Table 5. Specifically, in addition to the results presented in Table 1 and Table 2, we have added the results of modeling GuacaMol data with molecular graphs to better situate LO-ARMs++ in the literature. Note that, instead of only reporting one number for the GuacaMol benchmark, which is the standard reporting style in the literature, we report both means and standard deviations. For the baselines, we directly cite their reported results.

As we can see in both tables, although graph-based methods can yield best performance on novelty, probably due to an enlarged exploration space, their performance on all other distributional metrics lags behind SMILES-based methods with a significant margin. This suggests the supreme performance of SMILES-based methods, due to the simicility and efficiency of string representations.

LO-ARMs++ can not only surpass or match the performance of FO-ARMs on both benchmarks, but can also enable ARMs with the capability of learning the generation orderings by themselves.

## C GENERAL APPLICABILITY OF $\alpha$-$\beta$-ELBO

Although we developed LO-ARMs++ to target SMILES string generation, its core component, $\alpha$-$\beta$-ELBO, is actually modality-agnostic. To show the general applicability of $\alpha$-$\beta$-ELBO, we apply it to two challenging benchmarks of molecular graph generation, i.e., GuacaMol and ZINC250k graphs. The statistics of the two datasets are summarized in Table 6.

Especially, we reused the standard tokenization for molecular graph generation as well as the GraphTransformer, both of which are introduced in (Vignac et al., 2023). In this way, we can

Table 4: Molecule generation on GuacaMol SMILES dataset. We directly cite the results of other methods on the following metrics: **V**alidity, **U**niqueness, **N**ovelty, FCD and KL divergence. The metrics are calculated with the generated samples with the corresponding methods. In particular, the random sampler uniformly samples the test set.

| Method | Modality | V.%↑ | V.U.%↑ | V.U.N.%↑ | FCD↑ | KL Div.↑ |
|---|---|---|---|---|---|---|
| Random sampler | | 100.0 | 99.7 | 0.0 | 92.9 | 99.9 |
| DiGress | Graph | 85.2 | 85.2 | 85.1 | 68.0 | 92.9 |
| Cometh | Graph | 98.9 | 98.9 | 97.6 | 72.7 | 96.7 |
| DeFoG (50 sampling steps) | Graph | 91.7 | 91.7 | 91.2 | 57.9 | 92.3 |
| DeFoG (500 sampling steps) | Graph | **99.0** | **99.0** | **97.9** | 73.8 | 97.7 |
| AAE | SMILES | 82.2 | 82.2 | 88.0 | 52.9 | 88.6 |
| VAE | SMILES | 87.0 | 86.9 | 84.7 | 86.3 | 98.2 |
| LSTM ARM | SMILES | 95.9 | 95.9 | 87.4 | 91.3 | 99.1 |
| Our Results | | | | | | |
| AO-ARM | | $63.3 \pm 0.3$ | $63.2 \pm 0.3$ | $62.8 \pm 0.2$ | $72.1 \pm 0.7$ | $91.7 \pm 0.5$ |
| FO-ARM w/ standard tokenization | | $\mathbf{98.1 \pm 0.2}$ | $\mathbf{98.0 \pm 0.3}$ | $\mathbf{88.6 \pm 0.3}$ | $87.0 \pm 0.4$ | $99.1 \pm 0.1$ |
| FO-ARM w/ Prefix tokenization | | $83.3 \pm 0.7$ | $83.1 \pm 0.2$ | $82.8 \pm 0.3$ | $87.2 \pm 0.2$ | $99.1 \pm 0.2$ |
| LO-ARM | | $92.6 \pm 0.3$ | $92.6 \pm 0.3$ | $87.1 \pm 0.3$ | $79.4 \pm 0.3$ | $98.3 \pm 0.2$ |
| LO-ARM++ | | $93.9 \pm 0.2$ | $93.9 \pm 0.2$ | $85.9 \pm 0.3$ | $\mathbf{91.4 \pm 0.1}$ | $\mathbf{99.2 \pm 0.1}$ |

Table 5: Molecule generation on MOSES SMILES dataset. The metrics are the same as those in Table 2, and we directly cite the results reported by other baselines, including standard deviations when applicable. "-" means not applicable due to lack of reports.

| Method | Modality | Validity↑ | Novelty↑ | FCD↓ | SNN↑ | Frag↑ | Scaf↑ | IntDiv2↑ | logP↓ | SA↓ | QED↓ |
|---|---|---|---|---|---|---|---|---|---|---|---|
| Random sampler | - | 1.0 | 0.0 | 0.01 | 0.6419 | 1.0 | 0.991 | 0.851 | 0.0 | 0.0 | 0.0 |
| DiGress | Graph | 0.857 | 0.950 | 1.19 | 0.52 | - | 0.148 | - | - | - | |
| DisCo | Graph | 0.883 | **0.977** | 1.44 | 0.50 | - | 0.151 | - | - | - | |
| Cometh | Graph | 0.905 | 0.926 | 1.27 | 0.54 | - | 0.160 | - | - | - | |
| DeFog (50 sampling steps) | Graph | 0.839 | 0.969 | 1.87 | 0.50 | - | 0.235 | - | - | - | |
| DeFog (500 sampling steps) | Graph | 0.928 | 0.921 | 1.95 | 0.55 | - | 0.144 | - | - | - | |
| VAE | SMILES | **0.977** ± 0.001 | 0.695 ± 0.007 | 0.10 ± 0.01 | **0.626** ± 0.001 | 0.999 ± 0.000 | 0.939 ± 0.002 | 0.850 ± 0.000 | **0.023** - | 0.014 - | **0.0006** - |
| CharRNN | SMILES | 0.975 ± 0.026 | 0.842 ± 0.051 | **0.07** ± 0.02 | 0.602 ± 0.021 | **1.0** ± 0.0 | 0.924 ± 0.006 | 0.850 ± 0.001 | 0.057 - | 0.016 - | 0.0022 - |
| Our Results | | | | | | | | | | | |
| LO-ARM | SMILES | 0.663 ± 0.037 | **0.976** ± 0.013 | 4.39 ± 0.13 | 0.567 ± 0.010 | 0.954 ± 0.021 | 0.924 ± 0.003 | **0.857** ± 0.011 | 0.645 ± 0.006 | 0.18 ± 0.010 | 0.078 ±0.0006 |
| LO-ARM++ | SMILES | 0.946 ± 0.014 | 0.801 ± 0.021 | 0.14 ± 0.01 | 0.611 ± 0.000 | **1.0** ± 0.0 | **0.929** ± 0.003 | 0.854 ± 0.001 | 0.047 ± 0.002 | **0.012** ± 0.001 | 0.0015 ±0.0003 |

completely isolate the effect of $\alpha$-$\beta$-ELBO. Moreover, we reproduced the standard LO-ARMs (Wang et al., 2025a) through setting $\alpha = 1$ and $\beta = 0$ as constants in our LO-ARMs++ implementation.

Preliminary results show $\alpha$-$\beta$-ELBO improves validity and FCD on both benchmarks. The gain is more significant on GuacaMol, which is a more challenging dataset (max node number = 68) compared to ZINC250k (max node number=32). This suggests that curriculum learning with $\alpha$-$\beta$-ELBO is crucial for encouraging exploration and finding better local optima, especially in larger domains like GuacaMol graphs.

Note that we are yet to incorporate any graph-specific improvements (e.g., alternative Graph Transformer architectures (QIN et al., 2025)), which are complementary to the $\alpha$-$\beta$-ELBO.

Table 6: Ablation study on the standard and augmented tokenization algorithms
.

| Dataset | #samples | #nodes | #node types | Input dimensions |
|---|---|---|---|---|
| ZINC250k | 250k | $6 \leq V \leq 38$ | 9 | 1482 |
| GuacaMol Graphs | 1.2M | $2 \leq V \leq 63$ | 12 | 3906 |

Table 7: Preliminary results on molecular graph generation
.

| Dataset | Model | Validity%↑ | Uniqueness%↑ | Novelty%↑ | FCD↓ |
|---|---|---|---|---|---|
| ZINC205k | LO-ARM (Wang et al., 2025a) | $95.95 \pm 0.27$ | $100.0 \pm 0.0$ | $100.0 \pm 0.0$ | $3.33 \pm 0.13$ |
| | LO-ARM (reproduced) | $95.14 \pm 0.71$ | $100.0 \pm 0.0$ | $100.0 \pm 0.0$ | $3.49 \pm 0.10$ |
| | LO-ARM w/ $\alpha$-$\beta$-ELBO | $\mathbf{96.94 \pm 0.33}$ | $100.0 \pm 0.0$ | $100.0 \pm 0.0$ | $\mathbf{3.27 \pm 0.04}$ |
| GuacaMol | LO-ARM | $92.36 \pm 0.59$ | $100.0 \pm 0.0$ | $100.0 \pm 0.0$ | $5.21 \pm 0.06$ |
| | LO-ARM w/ $\alpha$-$\beta$-ELBO | $\mathbf{95.31 \pm 0.56}$ | $100.0 \pm 0.0$ | $100.0 \pm 0.0$ | $\mathbf{3.97 \pm 0.03}$ |

## D  ADDITIONAL ANALYSIS

### D.1  CONSISTENCY ANALYSIS FOR LEARNED GENERATION ORDERINGS

As the learned orderings with LO-ARMs++ are highly human-interpretable, to check the consistency of the learned orderings, we conducted the following steps:

- Step 1: Extract the pattern of each generation trajectory to a sequence of states. Specifically, `D` stands for digit, `A` for atom, and `P` for matching pair of parentheses. An example patten state sequence is `DDDDPPAAAAPAAA`.
- Step 2: Compress the state sequences through removing adjacent duplicates. For instance, for the example above, it is compressed to `DPAPA`.
- Step 3: Count the matchings of the following two templates: 1) first two states are `DP`, and 2) at least one `P` occurs after `DPA`.

### D.2  TRAINING INSTABILITY

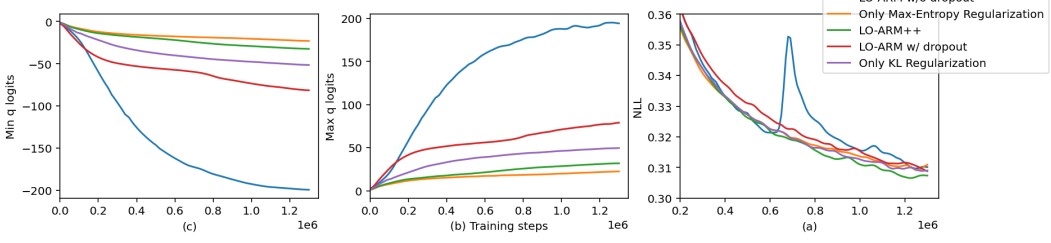

Figure 3: **Minimum (a) and maximum (b) logits outputted by** $q_\theta(z_i | z_{<i}, x)$ **and test negative log-likelihoods (NLLs) over the training course**. All the metrics evaluated against the test set.

We have observed two major issues that caused training instability.

First, we provide additional information about the evolution of the logit outputs of the variational order-policy $q_\theta(z_i | z_{<i}, x)$ along the training course. Specifically, Figure 3 (a) and (b) illustrate the minimum and maximum logit values in the outputs respectively. As we can see, the logits outputted by unregularized standard LO-ARM (blue curves) go to extremes quickly. In addition to the consequence of $q_\theta$ collapsing to premature orderings, such extreme values may also cause training instability, resulting in spikes in the test NLL (c). To fix this issue, we employed maximum entropy on the variational order-policy $q_\theta$, and we can see its effectiveness in Section 4.3.

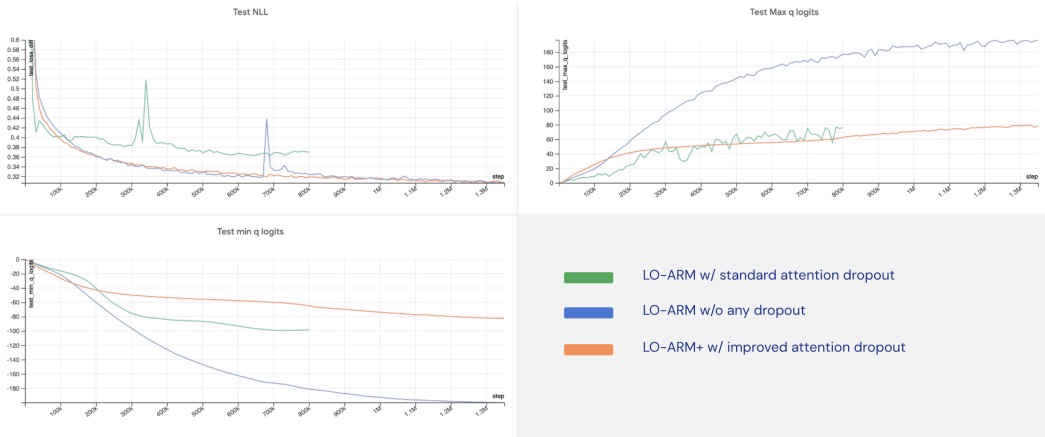

Figure 4: **Comparing different dropout methods applied to the generative model $p_\theta$.**. Specifically, 1) standard attention dropout (green), with which entries in attention score matrix are zeroed out directly, 2) improved attention dropout, with which we apply dropout to attention output (orange), and 3) no dropout (blue). All the metrics evaluated against the test set.

Second, another major source for training instability occurred when we applied standard attention dropout to the generative model $p_\theta$. As we can see in Figure 4, the standard attention dropout (green curves) resulted in large spikes in test NLL. We hypothesize that, because LO-ARMs model molecular strings of variable lengths and the padding dimensions are zeroed out in the attention score matrix, if we directly dropout the attention scores, the model would be confused with the dropped out dimensions and the padding dimensions, which are both zeros. Driven by this consideration, we change to apply dropout to the attention output, i.e, $\text{Attention}(Q, K, V) = \text{Dropout}(\text{softmax}\left(\frac{QK^T}{\sqrt{d_k}}\right) \cdot V)$. This simple yet effective fix yields stable generalization during training and improved generation performance at test time (orange curves).

Finally, one interesting observation is that, the instability occurred in the generative model $p_\theta$ also affects the variational order-policy $q_\theta$, as we can see the fluctuations in the test maximum $q$ logits in the green curve. This is because we are only using one optimizer to optimize these two networks, any instability in either network would be conveyed to the other through gradient backpropagation.

### D.3 ABLATION ANALYSIS ON PREFIX TOKENIZATION

We conduct an ablation study to compare two tokenization algorithms on the GuacaMol SMILES benchmark:

- Standard/character-based Tokenization: Parentheses are treated as individual tokens. This results in a vocabulary size of 109 after filtering.

- Prefix Tokenization: Pairs of parentheses are represented as single tokens. This leads to a vocabulary size of 129 after filtering.

For simplicity, we only run this ablation analysis for FO-ARM and LO-ARM (not LO-ARMs++), without incorporating the improvements introduced in this paper.

As shown in Table 8, for FO-ARMs (left-to-right generation order), the prefix tokenization downgrades validity while not affecting FCD. This is because the prefix tokenization not only increases the complexity of vocabulary, but also eliminates the left-to-right dependencies of matching parentheses, which yields a harder generation problem for FO-ARMs. On the other hand, as described in Appendix A.1, the prefix tokenization is a bijective transformation of SMILES strings, and therefore, not changing the effective data distribution, and the distribution of the valid molecules still matches with the ground truth data.

For LO-ARMs, as it is naturally capable of modeling non-left-to-right orderings, their FCDs are improved when using the prefix tokenization. We note that the validity of LO-ARM with standard

tokenization is decent (94.2%), while its FCD is much worse. This is because LO-ARM with standard tokenization tends to generate simple molecules without any ring structures.

Table 8: Ablation study on the standard and augmented tokenization algorithms
.

| Method | Tokenization | V.%↑ | V.U.%↑ | V.U.N%↑ | FCD↑ |
|--------|--------------|------|--------|---------|------|
| FO-ARM | Standard | $98.1 \pm 0.2$ | $98.0 \pm 0.3$ | $88.6 \pm 0.3$ | $87.0 \pm 0.4$ |
|        | Prefix | $83.3 \pm 0.7$ | $83.1 \pm 0.2$ | $82.8 \pm 0.3$ | $87.2 \pm 0.2$ |
| LO-ARM | Standard | $94.2 \pm 0.1$ | $94.0 \pm 0.1$ | $90.2 \pm 0.4$ | $36.6 \pm 0.2$ |
|        | Prefix | $92.6 \pm 0.3$ | $92.6 \pm 0.3$ | $87.1 \pm 0.3$ | $79.4 \pm 0.3$ |

### D.4 SENSITIVITY ANALYSIS ON ANNEALING HYPERPARAMETERS $\alpha$ AND $\beta$

In this analysis, we use the GuacaMol SMILES dataset as the proxy domain. For simplicity, we fix $\beta = 0.025$, only varying $\alpha$. In addition, we fix the exploration steps to 1M(illion) training steps.

Table 9: Sensitivity analysis of annealing hyperparameters
.

| $\alpha$ | #total training steps | FCD | Structure-first orderings | Has refinement steps |
|----------|----------------------|-----|---------------------------|----------------------|
| 1.025 | 2M | $91.3 \pm 0.10$ | Yes | Yes |
| 1.075 | 2M | $91.4 \pm 0.09$ | Yes | Yes |
| 1.125 | 2M | $91.4 \pm 0.08$ | Yes | Yes |
| 1.5 | 2M | $91.0 \pm 0.13$ | Yes | Yes |
| 1.5 | 3M | $91.2 \pm 0.11$ | Yes | Yes |

As is evident, when both the exploration steps and $\alpha$ are maintained within a suitable range, the performance exhibits a lack of sensitivity to their precise values, and the characteristic patterns of the learned orderings remain consistent. Specifically, the first three rows ($\alpha = 1.025, 1.075, 1.125$) unequivocally confirm the efficacy of the exploration-exploitation strategy fostered by the $\alpha$-$\beta$-ELBO. This strategy demonstrates two critical aspects: 1) The learning of efficient orderings through the exploration phase is paramount to achieving robust generation performance, and 2) once an efficient ordering is acquired, the model undergoes refinement via sufficient exploitation, which directly optimizes the unregularized ELBO ($\alpha$ progressively anneals to 1, and $\beta$ to 0).

Furthermore, sufficient exploitation effectively mitigates the sensitivity of the two hyperparameters, as evidenced by the cases where $\alpha = 1.5$ with total training steps of 2M and 3M. When the model is excessively regularized, the order policy tends toward uniformity, consequently introducing higher variance, and without adequate exploitation, the model may fail to converge optimally.

In conclusion, while we have introduced three supplementary hyperparameters, their impact on the final performance proves to be non-sensitive, provided that the model benefits from an adequate balance of exploration and exploitation.

## E EXPERIMENT SETUP

### E.1 MODELING SMILES STRINGS OF VARIABLE LENGTHS

The generative model $p_\theta$ and the variational distribution $q_\theta$, as shown in Equation (2), are both conditioned on the sequence length $L$. In practice, for a given SMILES string, the $L$ information is provided to both models via a sequence mask of a fixed maximum length (the maximum length across all ground truth data).

For the ground truth dataset (training, test, and validation sets), these sequence masks are generated directly from the actual data. Before sampling new molecules, however, we first sample the sequence length from a prior distribution, and then construct the corresponding sequence mask. During

inference, this sequence mask is fed to $p_\theta$ to distinguish between padding and the actual sequence dimensions.

## E.2 Model Architectures

The Transformer architecture is adopted from the `llama2.c` project[1]. For the FO-ARM model and the generative models $p_\theta$ in both LO-ARMs and LO-ARMs++, the corresponding Transformers consist of 18 attention layers. The variational order-policies used in LO-ARM and LO-ARMs++ have 6 attention layer. Moreover, We report the hyperparameters in Table 10. All experiments were run until convergence.

Table 10: Hyperparameter setup.

| Hyperparameter | ChEMBL/GuacaMol |
|---|---|
| Optimizer | AdamW |
| Scheduler | Cosine Annealing |
| Learning Rate | $5 \cdot 3^{-5}$ |
| Weight Decay | $1 \cdot 1^{-2}$ |
| EMA | 0.9999 |
| Attention dropout rate | 0.1 |
| Initial $\alpha$ | 0.025 |
| Terminating $\alpha$ | 0 |
| Initial $\beta$ | 1.05 |
| Terminating $\beta$ | 1 |
| Total training steps | $2e6$ |
| Exploration steps | $1e6$ |

## E.3 Annealing schedules for $\alpha$ and $\beta$

Our implementation utilizes a two-stage phased training strategy to balance exploration and exploitation:

- Exploration Stage: The KL regularization weight ($\beta$) is set to $\beta > 1$, and the maximum entropy weight ($\alpha$) is set to $\alpha > 0$.

- Exploitation Stage: These weights are fixed at $\beta = 1$ and $\alpha = 0$.

The total training duration is $2 \times 10^6$ steps. The Exploration Stage spans the first half of this duration, running for $1 \times 10^6$ steps.

Before training begins, the initial values are set to $\alpha = 0.025$ and $\beta = 1.05$. Throughout the Exploration Stage, both $\alpha$ and $\beta$ are annealed to their final termination values of $\alpha = 0$ and $\beta = 1$, respectively.

To ensure sufficient initial exploration, the annealing follows a two-part schedule:

- Persistent Stage: For the first half of the Exploration Stage ($500,000$ steps), both $\alpha$ and $\beta$ are held constant at their initial values.

- Linear Decay: Following the persistent stage, both weights undergo a linear decay, simultaneously reaching their termination values ($\alpha = 0$ and $\beta = 1$) exactly at the end of the full Exploration Stage.

We simplified the hyperparameter tuning by synchronizing the annealing schedules for $\alpha$ and $\beta$. The investigation of an asynchronized annealing approach is reserved for future work.

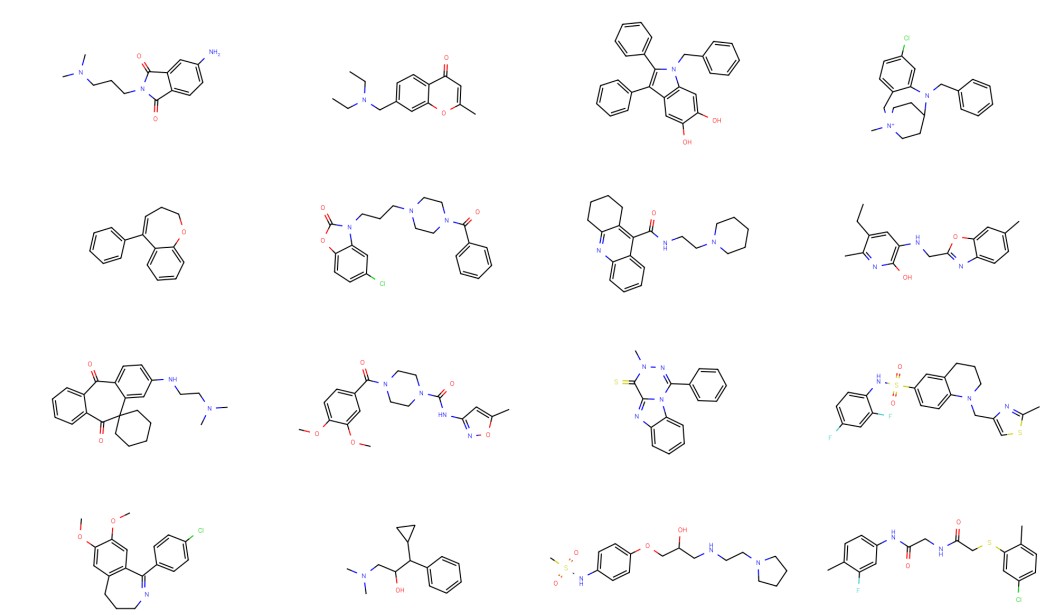

Figure 5: **Generated molecules with LO-ARMs++**.

## F    GALLERY OF GENERATED MOLECULES

## G    FULL STEP-WISE OUTPUTS FOR FIGURE 1

For each generation stage presented in Figure 1, we provide its full step-wise outputs in Figure 6 (Planning Stage), Figure 7 (Execution Stage), and Figure 8 (Refinement and Completion Stage). In addition to the partially generated SMILES strings and their corresponding partial 2D molecules (Column (a)), we also provide the outputs of the classifier (Column (b)) and the order-policy (Column (c)).

## H    AN ILLUSTRATION OF LEARNED SUB-OPTIMAL ORDER-POLICY

Figure 9 shows the process of generating a GuacaMol sample with a sub-optimal order-policy. Specifically, instead of proposing substructures at the initial stage, this generation process delays finalizing substructures to the very end there is no refinement stage with the sub-optimal order-policy. Therefore, this sub-optimal policy would be less generalizable to more complicated data distributions and would also be less tolerant to the generation errors in earlier steps.

---

[1]https://github.com/karpathy/llama2.c

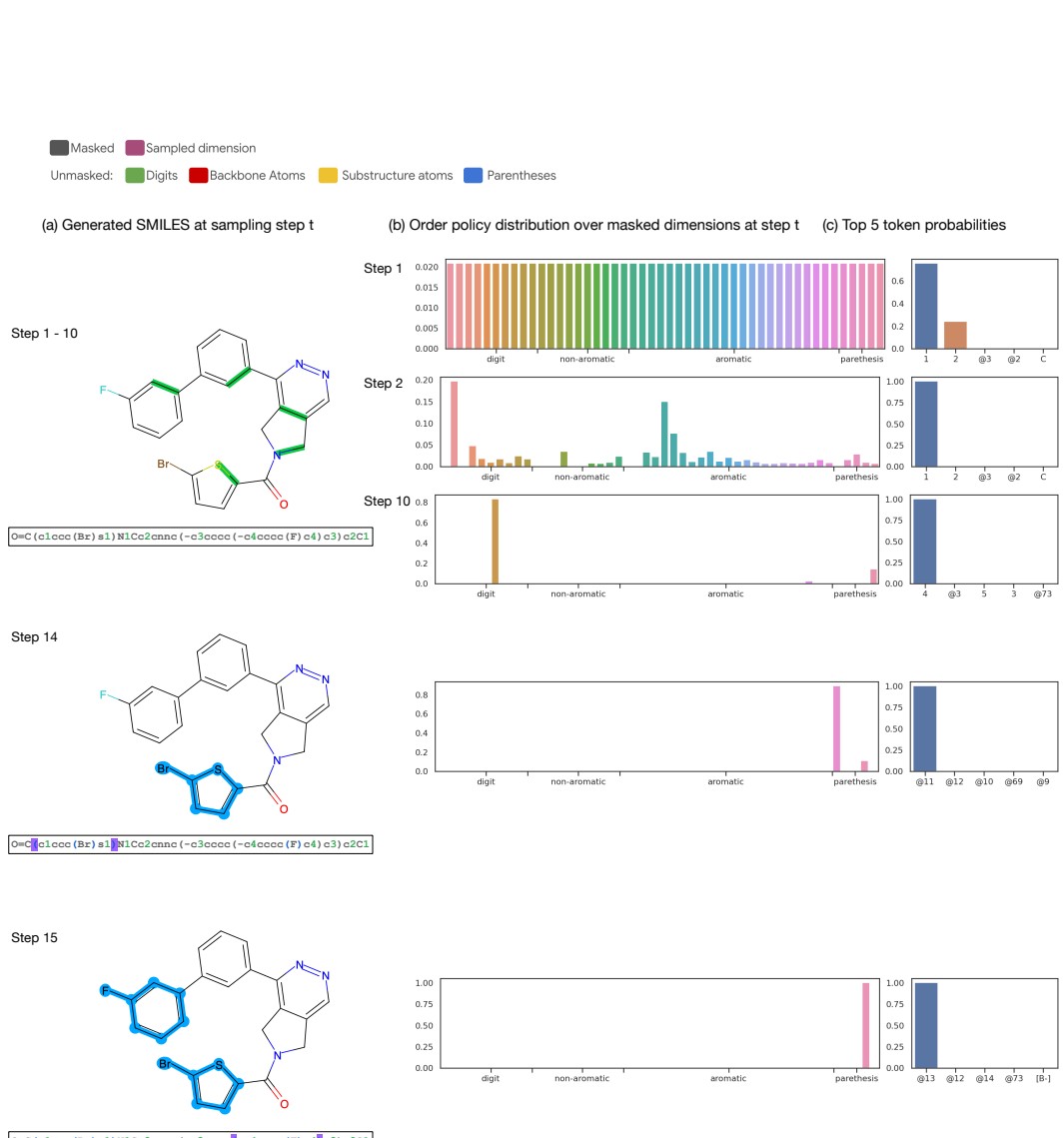

Figure 6: **Full step-wise outputs in the planning stage for Figure 1**. Our model generates SMILES strings step-by-step, commencing with all dimensions masked (in the figures masked dimensions are colored in grey) and adding token at a time. First, an *order-policy* selects which dimension to fill, and then a *classifier* determines its value. Each step is illustrated in the provided figures: Column (a) illustrate the (partially) generated SMILES string and the corresponding unmasked substructures in the final molecule (highlighted in colors). Columns (b) and (c) provide detailed insights: (b) the order-policy's probability distribution over dimensions, and (c) the classifier's prediction at the selected dimension. Note that, we only display the tokens of top 5 probabilities, and the order-policy is zeroed for unmasked dimensions. To facilitate visualization, we group the dimensions of the generated sample with respect to their dimension/token types: 1) digits (e.g., 1, 2), 2) non-aromatic tokens, (e.g., uppercase letters) 3) aromatic tokens (i.e., lowercase letters) and 4) parenthesis pairs. Notably, @N represents a pair of parentheses spanning $N$ dimensions between them.

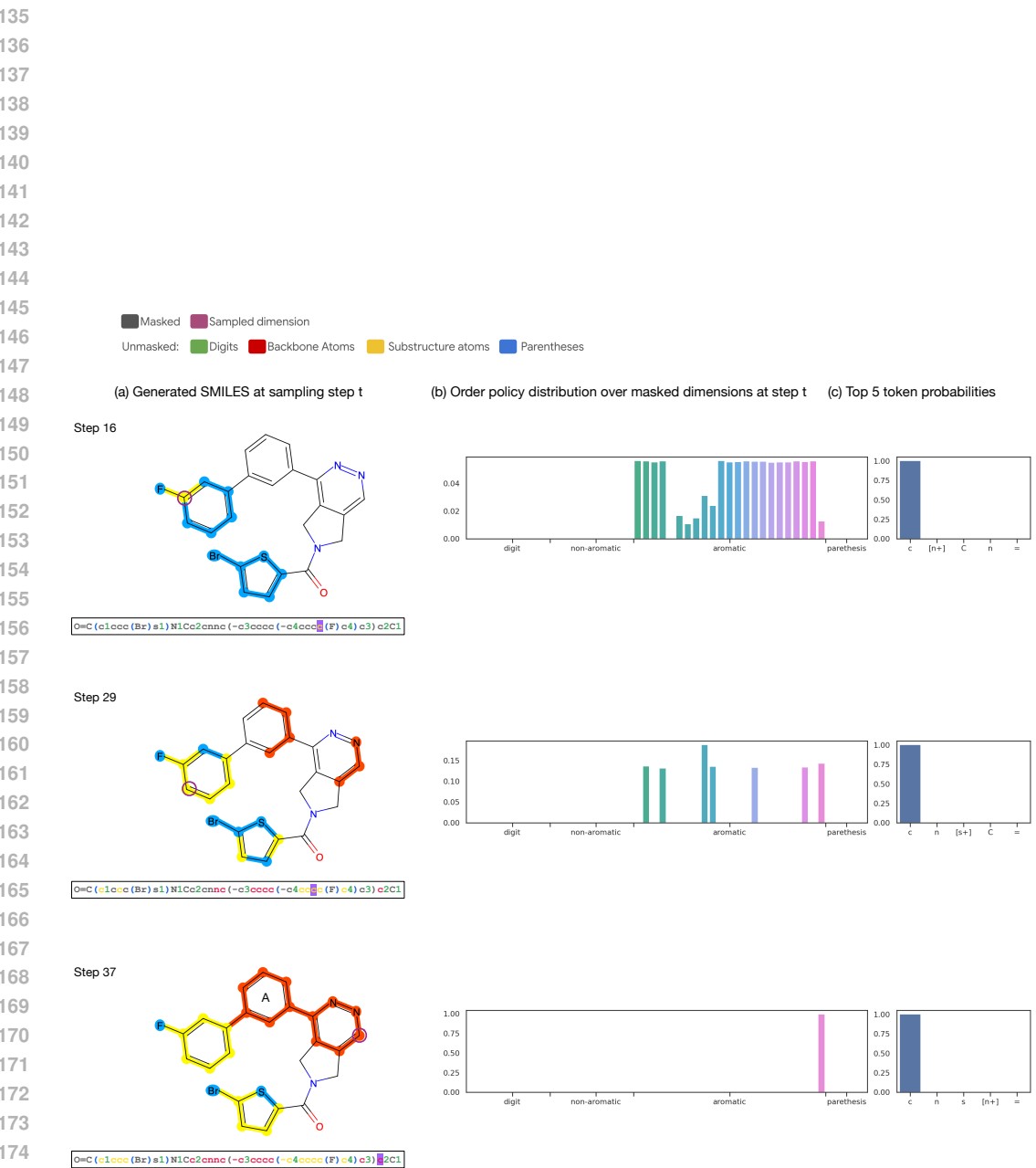

Figure 7: **Full step-wise outputs in the execution stage for Figure 1**. The legends in the bar plots are the same as those in Figure 6.

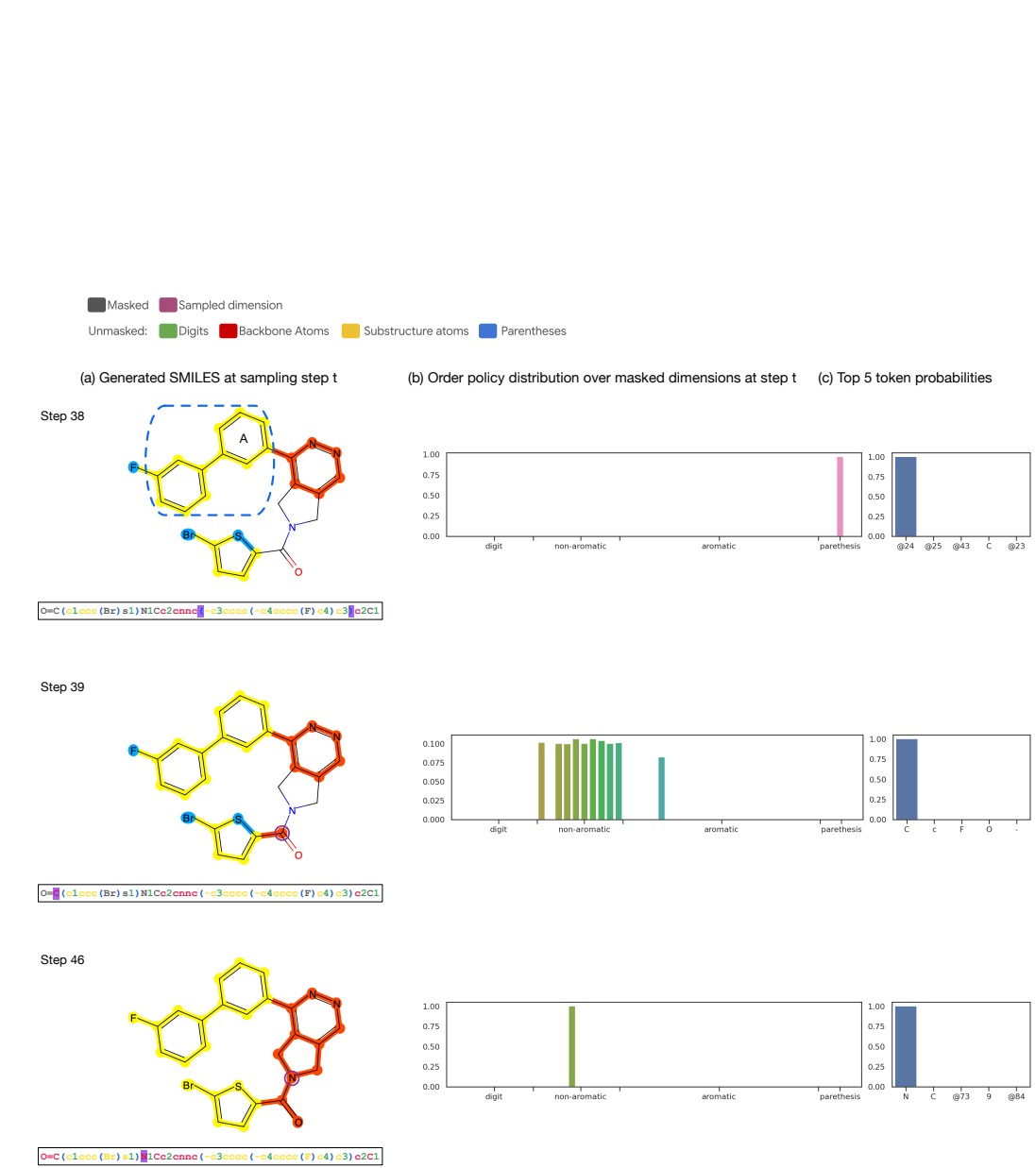

Figure 8: **Full step-wise outputs in the refinement and completion stage for Figure 1**. The legends in the bar plots are the same as those in Figure 6.

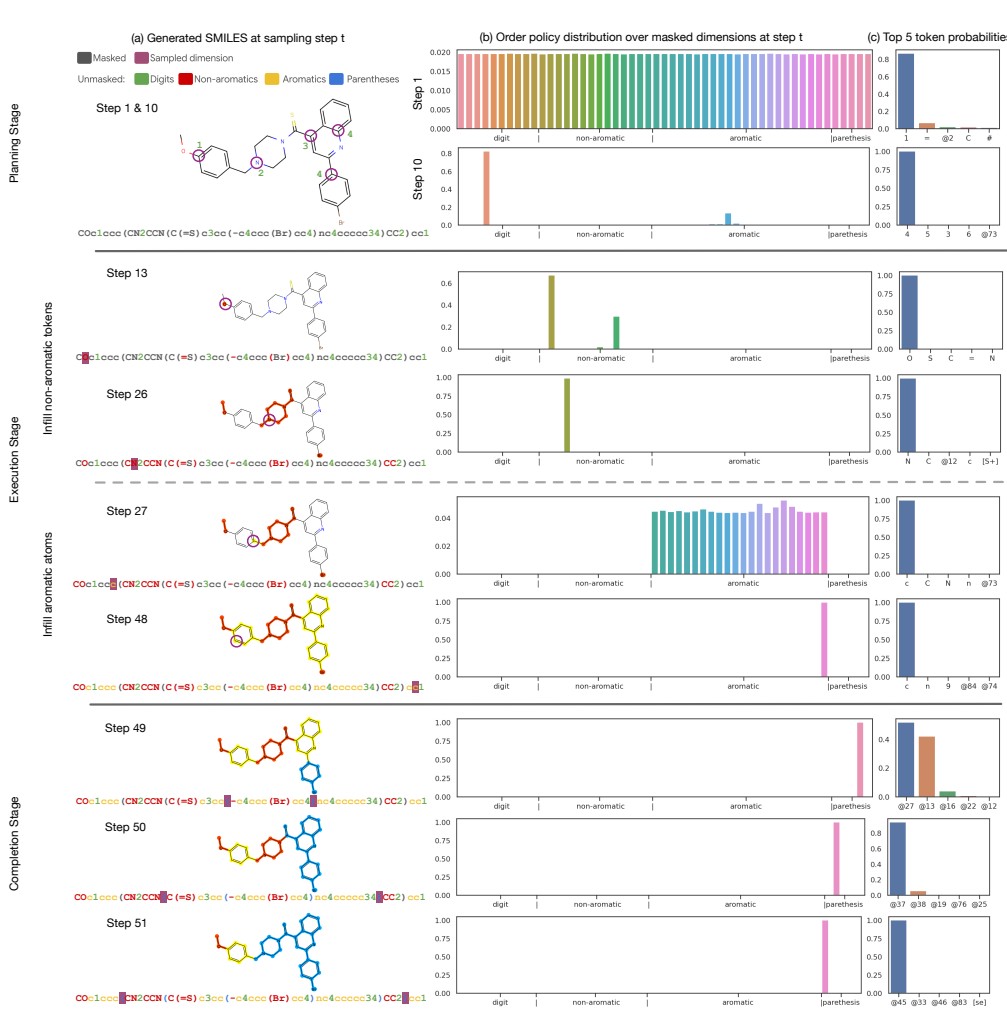

Figure 9: **An example of generating SMILES sample with a sub-optimal order-policy trained with the standard LO-ARM**. The legends in the bar plots are the same as those in Figure 6. The generation proceeds through three phases: 1) Planning (Step 1 to 10): LO-ARM first generates pairs of digits (highlighted in green), which represents ring closures. This step determines the number of rings and estimates their potential connections in the molecule. The digits together with their associated ring-cut atoms in the final sample are highlighted in the first molecule. 2) Execution (Step 13 to 48): The model then infills the molecular structure, characteristically generating non-aromatic tokens (red) before aromatic ones (yellow). 3) Completion (Step 49 to 51): Finally, it generates @N parenthesis tokens (blue) to enclose and finalize substructures.

