# OpenReview forum: "LO-ARMs++: Improving Learning-Order Autoregressive Models for Molecular String Generation"
_ICLR.cc/2026/Conference — Submitted to ICLR 2026_

### Official Review · Reviewer_BXeH · 2025-10-27

**Soundness:** 3
**Presentation:** 3
**Contribution:** 2
**Rating:** 2
**Confidence:** 4

**Summary:**

The paper presents LO-ARM++, an improvement over Learning-Order Autoregressive Models (LO-ARMs) for SMILES generation. The main contributions of the paper are:
- a tighter $\alpha$-$\beta$-ELBO training objective that adds maximum-entropy regularization on the variational order policy $q_\theta$.  $\alpha$ and $\beta$ are progressively decayed with a scheduler to implement an exploration-exploitation strategy to learn an optimal order policy.
- a stability fix that applies dropout to attention outputs instead of attention scores (which is sub-optimal for padded sequences).
- a prefix tokenization strategy that combines matched parenthesis pairs into single tokens.

LO-ARMS++ is tested on the GuacaMol benchmark, where it improves NLL and FCD over LO-ARM and performs comparably or better than fixed-order (FO) ARMs on distributional metrics (validity, uniqueness, novelty).

[1] Wang et al. Learning-order autoregressive models with application to molecular graph generation. ICML 2025

**Strengths:**

- The improvements over LO-ARM are tightly focused to resolve its shortcomings. Their choice is adequately motivated by ablations.
- The learned orderings are easy to interpret (although this was true already for LO-ARM).
- Performances are convincing (although not entirely transparent, see below).
- The paper is well written, easy to follow, with impacting visualizations.

**Weaknesses:**

- The paper reads like a well-engineered variant of LO-ARM rather than a new design or generative framework. In general terms, I'd positively value the contributions, but framed in ICLR's context, it has poor novelty value.
- The issue above is paired with the scope of the contributions, which is confined to one benchmark (GuacaMol), a single task (unconditional generation) and a single molecule tokenization (SMILES). While prefix tokenization is justifiable only for SMILES, it is not clear whether the rest of the methodology (in particular $\alpha$-$\beta$-ELBO and the dropout attention fix) is a genuine improvement that transcends the specific representation to which it is coupled. For example, would it work with SELFIES tokenization? Again, without wider validation, all of these sound like ad-hoc hacks tied to the specific data, task, and representation.
- Framing LO-ARM++ as a "discrete diffusion-style model" is a stretch (there is no diffusion in this model, only denoising). I suggest to either rephrase the claim, or to provide an adequate justification as to why the authors believe that's the case.
- If I understood correctly, low-frequency tokens (and the SMILES that contain them) are filtered out only for LO-ARM++ but not for the competitors, meaning that they are not trained on the same data, nor on the same representation (e.g. the vocabularies are different). This appers to be acknowledged in the paper, however the unfair setup makes the actual improvement brought forth by LO-ARM++ hard to judge.
- Although the impact of scheduling of $\alpha$ and $\beta$ is assessed, their default values are fixed withouth much justification. Could the authors provide a sensitivity analysis on these important parameters?

**Questions:**

See above.

---

> ### Author Response · Authors · 2025-11-24
>
> We thank the viewer for your advice and comments on our work, and we hope our response can fully resolve your concerns. We may rearrange the order of our responses to your concerns to make our response more cohesive.
>
> **[W: Weakness]**
>
> # W2: re more comprehensive evaluation
>
> We’ve provided extra evaluation on the following tasks, which makes our evaluation matrix more comprehensive, as shown in the Shared Responses 2, 3 and 4.
>
> Especially, LO-ARMs++ achieved SOTA performance on GuacaMol SMILES for distribution learning, surpassing LSTM, and outperformed CharRNN on 9 out of 12 distribution learning metrics on MOSES SMILES.
>
> To demonstrate the effectiveness of $\alpha$-$\beta$-ELBO, we applied it to two molecular graph generation tasks (ZINC250k, GuacaMol graphs) without modifying the tokenization or model architecture. The corresponding tables show that $\alpha$-$\beta$-ELBO improves both validity and FCD.
>
> # W1: concerns about the novelty of this work
>
> We respectfully disagree with the reviewer's novelty criticism, especially in the context of ICLR.
>
> We believe that our method is well-suited for ICLR. We promote representations via latent variables over deterministic latent embeddings, evidenced by the meaningful orderings learned by both LO-ARMs and LO-ARMs++. This paper addresses the challenge of training with variational inference in this context—specifically the local optima issue of the log-likelihood lower bound for LO-ARMs, which has a significant importance on model performance.
>
> While some elements are SMILES-specific, our main contribution, the $\alpha$-$\beta$-ELBO, is a general method for improving LO-ARMs' training stability and performance (similar to KL divergence warm-up in VAEs [1]). This effective and scalable regularization improves the training of LO-ARMs and yields more sample-efficient learned orderings in both SMILES and molecular graph generation tasks.
>
> # W4: Re the effect of filtering low-frequency tokens
>
> Our evaluation uses the unfiltered data as the reference for a fair comparison. Filtering only controls model size by removing low-frequency tokens, mostly large parenthesis pairs. As shown in Table 2, we filtered less than 1%, which minimally impacts the training set distribution.
>
> # W3: Connection between LO-ARMs++ and diffusion models
>
> We referred to the connection of LO-ARMs++ to masked diffusion models. Masked diffusion models are equivalent to Any-Order ARMs (AO-ARMs) [2], which are a special case of LO-ARMs where the two order policies (p and q) are fixed uniform distributions. Thus, LO-ARMs generalize AO-ARMs by making the order policies learnable. Specifically, instead of having a fixed forward process (uniform in the case of AO-ARMs), the variational order policy $q$ is actually a learnable forward process.
>
> # W5: Sensitivity analysis of $\alpha$ and $\beta$
>
> We are currently completing this analysis.. Hope this would fully address your question.
>
> [1] Sønderby, C. K., Raiko, T., Maaløe, L., Sønderby, S. K., & Winther, O. (2016). Ladder variational autoencoders. Advances in neural information processing systems, 29.
>
> [2] https://arxiv.org/pdf/2509.01025

---

> > ### Comment · Reviewer_BXeH · 2025-11-25
> > **Thank you**
> >
> > I thank the authors for their detailed rebuttal and revisions, and I have slightly increased my score to acknowledge these efforts. Nonetheless, my primary concern remains: the paper still appears largely as an improved re-engineering of LO-ARMS, addressing some of its shortcomings but without, in my opinion, a substantial conceptual novelty. To be clear, I find the paper of good quality, but I am not entirely convinced it reaches the typical ICLR bar in terms of novelty. That said, I would not oppose acceptance if the following discussion with the AC and the other reviewers will suggest acceptance.

---

> ### Author Response · Authors · 2025-11-28
> **Thank you for your swift and constructive feedback**
>
> We thank you for your swift and constructive response as well as your acknowledgement of the contributions and quality of our work. In these two responses, we will first address your question about the sensitivity analysis of the hyperparameters $\alpha$ and $\beta$, and then we’ll address your concern about the novelty of our work in the next response.
>
> # W5 cont'd: Sensitivity analysis of $\alpha$ and $\beta$
>
> Thank you for this great question. In this analysis, we use the GuacaMol SMILES dataset as the proxy domain. For simplicity, we fix $\beta = 0.025$, only varying $\alpha$. In addition, we fix the exploration steps to 1M(illion) training steps.
>
> | $\alpha$ | Total training steps | FCD         | Structure-first orderings | Has refinement steps |
> |----------|------------|-------------|-----------------|-----------------|
> | 1.025    | 2M           | 91.3 $\pm$ 0.10 | Yes             | Yes                  |
> | 1.075 (reported in Response 2)    | 2M           | 91.4 $\pm$ 0.09 | Yes             | Yes                  |
> | 1.125    | 2M           | 91.4 $\pm$ 0.08 | Yes             | Yes                  |
> | 1.5      | 2M             | 91.0 $\pm$ 0.13 | Yes             | Yes                  |
> | 1.5      | 3M             | 91.2 $\pm$ 0.11 | Yes             | Yes                  |
>
> As is evident, when both the exploration steps and $\alpha$ are maintained within a suitable range, the performance exhibits a lack of sensitivity to their precise values, and the characteristic patterns of the learned orderings remain consistent. Specifically, the first three rows ($\alpha=1.025, 1.075, 1.125$) unequivocally confirm the efficacy of the exploration-exploitation strategy fostered by the $\alpha$-$\beta$-ELBO. This strategy demonstrates two critical aspects: 1) The learning of efficient orderings through the exploration phase is paramount to achieving robust generation performance, and 2) once an efficient ordering is acquired, the model undergoes refinement via sufficient exploitation, which directly optimizes the unregularized ELBO ($\alpha$ progressively anneals to 1, and $\beta$ to 0).
>
> Furthermore, sufficient exploitation effectively mitigates the sensitivity of the two hyperparameters, as evidenced by the cases where $\alpha=1.5$ with total training steps of 2M and 3M. When the model is excessively regularized, the order policy tends toward uniformity, consequently introducing higher variance, and without adequate exploitation, the model may fail to converge optimally.
>
> In conclusion, while we have introduced three supplementary hyperparameters, their impact on the final performance proves to be non-sensitive, provided that the model benefits from an adequate balance of exploration and exploitation.
>
> We believe this completes our responses to all your technical questions. Please don't hesitate to let us know if you have any other questions.

---

> ### Author Response · Authors · 2025-12-04
>
> Thank you for your feedback.
>
> The ML community tends to favor straightforward and efficient methods over intricate ones, as simple yet effective approaches can be rapidly adopted and easily generalized across various research domains, suggesting a clear preference for simplicity and widespread applicability.  This is also our philosophy of developing LO-ARMs++, only reserving the most effective changes without complicating our methods.
>
> On a note of conceptual novelty, in addition to demonstrating it through the simplicity and effectiveness of our proposed methods, achieving SOTA results on the GuacaMol benchmark together with the enhanced orderings learned with LO-ARMs++, we grounded our development in variational inference and introduced the Generalized Next-Token-Predictor (NTP), connecting fixed-order (FO), any-order (AO) and learning-order (LO) autoregressive models (ARMs) under a unified variational inference framework, which could be beneficial to the wider ML community.

---

### Official Review · Reviewer_WFFz · 2025-10-29

**Soundness:** 3
**Presentation:** 3
**Contribution:** 3
**Rating:** 6
**Confidence:** 3

**Summary:**

This paper proposes LO-ARMs++, an improved training strategy for Learning-Order Autoregressive Models in SMILES-based molecular generation. The core idea is an α–β–ELBO objective that jointly enforces (1) maximum-entropy exploration on the variational order policy and (2) KL distillation from the variational policy into the model policy — with annealing schedules to gradually shift from exploration to exploitation. Two additional engineering contributions — prefix-style bracket tokenization and attention-output dropout — stabilize training. On GuacaMol distribution matching, LO-ARMs++ closes most of the gap to strong fixed-order models and even surpasses them on FCD, while producing interpretable “structure-first” generation orders.

**Strengths:**

- Clear motivation addressing a known pathology of LO-ARM — early collapse of the order distribution.

- Conceptually simple yet effective α–β–ELBO, with solid grounding in max-entropy variational inference / diffusion-like frameworks.

- Strong empirical gains, especially over vanilla LO-ARM, matching or exceeding fixed-order baselines on GuacaMol.

- Thorough ablations, showing the necessity of both α (entropy) and β (distillation) plus the improved dropout.

- Interpretability results are concrete and quantified — the model reliably learns “plan → refine” generation order.

- Engineering contributions are practical and reusable beyond chemistry (tokenization + attention dropout fix).

**Weaknesses:**

1. Single benchmark focus — experiments are limited to GuacaMol distribution-learning only (no goal-directed tasks or additional datasets).

2. Likelihood claims are partially qualified — FO vs LO NLLs are not fully comparable due to different length-conditioning setups.

3. No variance/CI reporting — results are averaged over 5 seeds but lack dispersion or statistical uncertainty.

4. Length prior is assumed rather than studied — impact on generation quality is unclear.

5. Some claims (e.g., “first diffusion-style model to reach this level”) may be slightly overstated without broader comparisons.

**Questions:**

1. Can you report mean ± std or confidence intervals for the main table, to quantify robustness?

2. How is the length prior estimated / smoothed, and what is its effect on FCD / uniqueness / novelty?

3. Have you tested LO-ARMs++ on goal-directed GuacaMol tasks, or any dataset beyond SMILES distribution modeling?

4. Could an asynchronous schedule for α and β be even better (e.g., entropy decays slower than distillation)?

5. Can you release code, especially for the bracket-prefix tokenizer and attention-output dropout patch?

---

> ### Author Response · Authors · 2025-11-24
>
> We sincerely thank you for your positive feedback, and we are addressing your comments below.
>
> **[W: Weakness, Q: Question]**
>
> # W1 & Q3: re limited evaluation on single benchmark.
>
> We’ve improved the performance of LO-ARMs on GuacaMol SMILES and have  provided extra evaluation, which makes our evaluation matrix more comprehensive, as shown in the shared responses (i.e., Response 2, 3 and 4).
>
> Especially, LO-ARMs++ achieved SOTA performance on GuacaMol SMILES for distribution learning, surpassing LSTM, and outperformed CharRNN on 9 out of 12 distribution learning metrics on MOSES SMILES.
>
> To demonstrate the effectiveness of $\alpha$-$\beta$-ELBO, we applied it to two molecular graph generation tasks (ZINC250k, GuacaMol graphs) without modifying the tokenization or model architecture. The corresponding tables show that $\alpha$-$\beta$-ELBO improves both validity and FCD.
>
> # W3 & Q1: missing CI/stddev in reporting.
>
> We have updated the LO-ARMs++ results in the centralized response tables to include standard deviation (stddev). See Shared Response 2 for extra information on the issues of standard evaluation. We cite reported results for literature baselines (no stddev). Baselines we implemented will be fully updated by the end of the rebuttal period.
>
> # W2: Inappropriate comparison in NLLs between FO-ARMs and LO-ARMs
>
> We agree that NLLs may not be the most appropriate metric to report due to differences in inputs (LO-ARMs use additional sequence masks). Therefore, we prioritize molecule generation distribution learning metrics and have removed NLLs from the revised tables in Shared Response 2.
>
> In addition, following the advice of Reviewer iSaf, we are rerunning experiments of FO-ARMs to exclude the paddings in NLL calculations, and we will update the results as soon as possible.
>
> # W4 & Q2: re effect of sequence length on generation performance
>
> Thank you for the advice. We use the distribution of raw training set sequence lengths as the prior distribution, without filtering or smoothing. This prior is only used during inference: we first sample the prior to obtain a batch of sequence lengths, use them to construct sequence masks, and then feed these masks to the generative model (conditioning on L as in Eq(1)).
>
> We are currently completing an analysis to investigate the effect of fixing the sequence length within a batch on generation performance (validity, uniqueness, FCD, KL divergence). We hope this addresses your question.
>
> # Q4: re asynchronous annealing schedule for $\alpha$-$\beta$-ELBO.
>
> Thank you. We agree that slower entropy decay than distillation would be beneficial, and this is implicitly the default. In Equation (5), the effective entropy decay is $\alpha + \beta$, where the max entropy regularization is always larger than that of the cross entropy term. This may have the same effect as longer entropy decay. This implementation simplifies hyperparameter configuration by avoiding an extra parameter for $\alpha$ and $\beta$ decay steps.
>
> This configuration is effective for SMILES generation, but we acknowledge that explicitly implementing an asynchronous annealing schedule would be more flexible for future work.
>
> # W5: re possible overstatement about the comparison to masked diffusion
>
> Thank you very much for your advice. As you can see in our extended evaluation, we’ve also evaluated on the MOSES dataset in addition to the GuacaMol SMILES dataset, which are two major large-scale molecular generation benchmarks. In both benchmarks, LO-ARMs++ have achieved decent or SOTA results.
>
> We’ll rephrase the statement in the updated PDF, to be specific to the benchmarks of MOSES and GuacaMol SMILES.
>
> Moreover, in the experiments of molecular graph generation, we can also see that $\alpha$-$\beta$-ELBO is generally efficient in improving generation performance, especially FCD and validity.
>
> # Q5: open sourcing code
>
> Of course! We’ll prepare for open sourcing our code upon the acceptance of this paper. It’ll be our pleasure and great honor to see our work to be adopted in the wider community.

---

> ### Author Response · Authors · 2025-11-26
> **Cont'd: Addressing Q2: re the effect of length prior on generation performance**
>
> We thank the reviewer for your advice. To investigate how the the length prior affects the generation performance of LO-ARMs++, we ran the following analysis on the GuacaMol SMILES dataset, the setup of which is as below:
>
> 1) Instead of sampling the prior length distribution, we fix the sequence length for a batch, systematically varying it from 20 to 120. These lengths correspond to the sequence after prefix tokenization, for which we also report the corresponding average length of the detokenized SMILES.
>
> 2) We further include the percentage of molecules corresponding to each fixed length within the training set, illustrating the extent to which the generated samples deviate from the prior length distribution.
>
> 3) Consistent with the refined evaluation protocol, we report the metrics for Validity (V), Validity & Uniqueness (V&U), Validity, Uniqueness, & Novelty (V&U&N), and the FCD score. Crucially, for all FCD scores, the test set is employed as the reference dataset.
>
> N.B., the sequence length for the training set, following the application of prefix tokenization, exhibits an average of 46.5 tokens, with a maximum length recorded at 96 tokens.
>
> | Fixed length                          | 20          | 40          | 60          | 80          | 100         | 120         |
> |---------------------------------------|-------------|-------------|-------------|-------------|-------------|-------------|
> | Molecule proportion in training set   | 0.5%        | 3.4%        | 0.8%        | 0.1%        | 0.          | 0.          |
> | actual avg. generated sequence length | 22.6 $\pm$ 1.3  | 45.3 $\pm$ 3.2  | 68.7 $\pm$ 2.3  | 92.4 $\pm$ 4.9  | 115.0 $\pm$ 7.3 | 137.4 $\pm$ 8.1 |
> | Validity %                            | 92.0 $\pm$ 0.9  | 94.7 $\pm$ 0.2  | 92.3 $\pm$ 0.4  | 81.1 $\pm$ 0.5  | 60.1 $\pm$ 0.3  | 21.6 $\pm$ 0.9   |
> | V.U %                                 | 81.2 $\pm$ 0.8  | 97.5 $\pm$ 0.7  | 94.4 $\pm$ 0.4  | 92.9 $\pm$ 0.8  | 100.0 $\pm$ 0.0 | 100.0 $\pm$ 0.0 |
> | V.U.N %                               | 84.7 $\pm$ 0.3  | 87.3 $\pm$ 0.1  | 92.1 $\pm$ 0.1  | 98.2 $\pm$ 0.2  | 100.0 $\pm$ 0.0 | 100.0 $\pm$ 0.0 |
> | FCD score %                           | 7.53 $\pm$ 0.12 | 77.1 $\pm$ 0.37 | 31.9 $\pm$ 0.32 | 3.61 $\pm$ 0.05 | 0.45 $\pm$ 0.02 | 0.15 $\pm$ 0.01 |
>
> As the table clearly demonstrates, when the fixed sequence lengths remain within the bounds of the prior distribution, LO-ARMs++ successfully generates molecules exhibiting satisfactory validity, uniqueness, and novelty, even in the distribution's tail (e.g., length=80, which represents 0.1% of the molecules). Nonetheless, as the sample length distribution increasingly deviates from the prior, a commensurate degradation in both validity and FCD is observed, aligning with expectations.
>
> LO-ARMs++ was stress-tested with out-of-distribution (OOD) generation by fixing sequence lengths to 100 and 120 (lengths absent from training data). Despite these demanding OOD conditions, LO-ARMs++ successfully generated valid molecules with 100% uniqueness and novelty. It would be infeasible to generate such OOD samples with FO-ARMs because they lack sequence length conditioning.
>
> We believe we have fully addressed your concerns, justifying a more favorable assessment. Please don’t hesitate to let us know if you have any questions.

---

### Official Review · Reviewer_cn57 · 2025-10-31

**Soundness:** 1
**Presentation:** 3
**Contribution:** 1
**Rating:** 2
**Confidence:** 3

**Summary:**

This work proposes a set of modifications to an existing discrete diffusion-based model for generating molecules.
Concretely, changes to tokenization, loss function and regularisation are the main focus points;
Experiments on the GuacaMol benchmark suggest favourable comparison compared to the base model (without modifications).

**Strengths:**

- Molecule generation is a relevant task, given the number of papers written on the topic.
 - The proposed modifications have not been published elsewhere.
 - Overall, the paper is clearly structured and well-written.

**Weaknesses:**

- The results in table&nbsp;1 indicate that the method is not competitve with existing methods.
   First of all, the way novelty is measured is not very meaningful and can be trivially increased (Renz et al., 2019).
   Better metrics for evaluating space exploration have been provided (e.g. Xie et al., 2023).
   On the other metrics, the model does not manage to match results from an LSTM model from 2019.
   Finally, the comparison ignores relevant modern molecule generation models that present even better results (e.g. Özçelik et al., 2024)
 - The proposed modifications are marginal at best.
   The prefix tokenization is a clean solution to the challenge with parentheses, but is limited to adding one special token.
   Similarly, the dropout is a nice workaround for a practical problem, but comes down to changing the order of layers slightly.
   Finally, the proposed $\alpha$-$\beta$-ELBO comes down to adding tunable scaling factors in the loss function.
   None of these modifications provide significant(ly new) insights.
   Furthermore, these modifications are very much tailored to LO-ARMs,
   making their general applicability questionable.
   Because these changes do not suffice to make LO-ARMs competitive with modern baselines,
   I fail to find much value for the general ML community.
 - It is not entirely clear why it would be necessary or beneficial to allow different orderings.
   After all, this work seems to report results that rely on canonicalized SMILES, which should have a fixed ordering anyway.
   On the other hand, even if the model is able to produce non-canonical SMILES, it does not seem to provide any advantage.
 - It is unclear whether the SMILES are canonicalised during training.
   On line 331 the authors suggest that the model is able to learn custom orderings,
   suggesting that the model has seen non-canonical SMILES during training.
   However, this does not seem to be mentioned explicitly.

### Minor Issues
 - AO-ARM acronym is used before the acronym (or the corresponding model) is introduced (line 219).

### Additional References
 - Özçelik et al. (2024). [Chemical language modeling with structured state space sequence models](https://www.nature.com/articles/s41467-024-50469-9). Nature Communications, 15(1), 6176.
 - Renz et al. (2019). [On failure modes in molecule generation and optimization](https://doi.org/10.1016/j.ddtec.2020.09.003). Drug Discovery Today: Technologies, 32, 55-63.
 - Xie et al. (2023) [How Much Space Has Been Explored? Measuring the Chemical Space Covered by Databases and Machine-Generated Molecules](https://openreview.net/forum?id=Yo06F8kfMa1). International Conference on Learning Representations.

**Questions:**

1. How does this method perform on the #circles metric (Xie et al., 2023) in comparison to other models?
 2. How does the method perform compared to state-of-the-art molecule generation methods?
 3. Are there any (new) insights in this work that could be useful for the more general ML community?
 4. Were all models trained on canonicalized SMILES? If not, which models were trained with arbitrary SMILES?
 5. What is the value of a model that is not sensitive to the ordering of SMILES, given canonical SMILES?
 6. How does the prefix tokenization affect other baseline models (e.g. LSTM)?

---

> ### Author Response · Authors · 2025-11-24
> **Part I**
>
> We sincerely thank the reviewer for your thorough review as well as sharing your deep knowledge in the chemistry domain. In this response, we rearrange the order of our responses to the corresponding weaknesses and questions to make our presentation more coherent. We hope that our response can fully address your concerns.
>
> **[W: Weakness, Q: Question]**
>
> # W1.B & Q2: Comparing to SOTA molecule generation methods
>
> We have enhanced our model, with LO-ARMs++ now achieving SOTA performance on the GuacaMol SMILES benchmark. This performance surpasses LSTM in SMILES generation and significantly outperforms other graph-based methods (see Response 2). Furthermore, on the less complex MOSES benchmark, LO-ARMs++ outperforms CharRNN/LSTM across nine out of twelve distribution learning metrics, as detailed in Response 3.
>
> # W1.A & Q1 Reporting on a more comprehensive set of metrics & evaluation against the #circles metric.
>
> In our updated presentation, we have adhered to standard literature reporting by including both FCD and KL divergence for the GuacaMol benchmark, and supplementary chemistry metrics, such as SNN, logP, and QED, for the MOSES benchmark.
> We are currently fixing a bug with the #Circle metric library and will report results as soon as possible. Due to time constraints and the lack of other baseline data, our comparison in this rebuttal will be restricted to LO-ARMs++ versus the ground truth dataset for the #Circles metric.
>
> # W1.C Missing essential reference on the latest results
>
> Thank you for sharing Özçelik et al. (2024). We seek further advice on these points:
>
> 1. Our GuacaMol dataset uses ChEMBL v24, while Özçelik et al. (2024) uses ChEMBL v31. Since the datasets and preprocessing may differ, comparing results is infeasible [1].
>
> 2. Our research prioritizes distribution learning (the ability to sample new molecules from the same chemical space). Özçelik et al. (2024) did not report FCD (a distribution learning metric) on the ChEMBL benchmark, making comparison difficult.
>
> 3. On the MOSES benchmark, Özçelik et al. (2024) used 102,400 SMILES strings for evaluation, which is fine for validity, novelty, and uniqueness. However, this sample size is unfair for FCD comparison; standard MOSES evaluation uses only 30,000 valid samples [2].
>
> # W2.A & W2.B Simplicity and effectiveness of proposed methods and their contributions and value for the ML community
>
> We respectfully disagree with the characterization of our proposed modifications as marginal. While certain elements, such as the prefix tokenization and the attention-output dropout patch, are specifically tailored for the SMILES generation task, our primary contribution, the $\alpha$-$\beta$-ELBO, is a fundamentally general and essential method for improving the training stability and performance of LO-ARMs. This approach draws parallels to established techniques like the introduction of the KL divergence warm-up parameter in VAEs [5]. Not only is our method effective, but it’s also scalable as we’ve already seen in the evaluation of both SMILES and molecular graph generation tasks.
>
> Moreover, LO-ARMs++'s improvements—prefix tokenization and attention-output dropout—complement $\\alpha$-$\\beta$-ELBO. As Reviewer WFFz noted, these would be valuable for the research community. We will open-source our code upon acceptance.
>
> [1] https://github.com/BenevolentAI/guacamol/blob/master/guacamol/data/get_data.py
>
> [2] https://arxiv.org/abs/1811.12823
>
> [3] https://github.com/molecularsets/moses/tree/master/data
>
> [4] https://github.com/cvignac/DiGress/tree/main/src/datasets
>
> [5] Sønderby, C. K., Raiko, T., Maaløe, L., Sønderby, S. K., & Winther, O. (2016). Ladder variational autoencoders. Advances in neural information processing systems, 29.

---

> ### Author Response · Authors · 2025-11-24
> **Part II**
>
> # W3 & Q5: Impact of learning generation ordering
>
> We thank you for your question. It's crucial to distinguish between token orderings based on the SMILES grammar (observed from data) and the generation orderings learned by the model (representing latent dependencies). Canonical SMILES, while efficient for raw data, may not effectively capture hidden dependencies (e.g., matching ring closure numbers like '1' in
>
> COc1cccnc1COc1ccc(NC(=O)Nc2ccc3cccc(OCCCC(=O)O)c3c2)cc1).
>
> Our work aims to learn generation orderings to uncover chemically plausible, unobserved token dependencies. As shown in Figure 1, the learned orderings differ from Depth-First-Search (DFS)-based SMILES generation. LO-ARMs++ initially proposes molecular structures, then infills them with atom tokens, dynamically refining the structure during generation. This learned ordering would be valuable for structure-constrained molecule design, an application we defer to future work, as this paper focuses on improving LO-ARMs training.
>
> # W4 & Q4: Clarification on datasets
>
> For all raw datasets used in this research, we directly use the standard ones provided in the literature to be consistent with corresponding baselines. Specifically,
>
> 1) for GuacaMol, canonicalizing SMILES seems a standard step in preprocessing, which is not only employed in the GuacaMol dataset, but also used in Özçelik et al. (2024). So we also followed this standard on the GuacaMol benchmark. All our models were trained on the canonicalized SMILES, as well as other baselines in the literature.
>
> 2) For the MOSES dataset, we directly used the raw and non-canonicalized dataset provided in [3].
>
> 3) For molecular graphs, we used the library provided in [4] to prepare our data.
>
> # Q6: Effect of the prefix tokenization on other baselines.
>
> We’re preparing the corresponding ablation study, and will provide the analysis soon. Thank you for your patience.
>
>
> [1] https://github.com/BenevolentAI/guacamol/blob/master/guacamol/data/get_data.py
>
> [2] https://arxiv.org/abs/1811.12823
>
> [3] https://github.com/molecularsets/moses/tree/master/data
>
> [4] https://github.com/cvignac/DiGress/tree/main/src/datasets
>
> [5] Sønderby, C. K., Raiko, T., Maaløe, L., Sønderby, S. K., & Winther, O. (2016). Ladder variational autoencoders. Advances in neural information processing systems, 29.

---

> ### Comment · Reviewer_cn57 · 2025-11-26
> **Thank you for addressing my concerns**
>
> > your deep knowledge in the chemistry domain
>
> I am not sure what you want to imply with this statement, but my expertise is definitely not in chemistry
>
> > We have enhanced our model, with LO-ARMs++ now achieving SOTA performance
>
> Response 2 seems to suggest that the number of parameters in the generator was unchanged to maintain the same capacity. However, does this also mean that the LSTM and LO-ARMS++ have a comparable number of parameters? I could not find this information on a quick search in the manuscript.
>
> I do not understand the implications of response 3. It shows that the CharRNN is better on the metrics that are reported for most results in the manuscript (validity, novelty and FCD), but now there is a collection of new metrics where the LO-ARMS++ performs better? Furthermore, there are no bold figures in the CharRNN row of the table and most of these new metrics are not very meaningful (e.g. see Renz et al., 2024, figure 1). I believe this table is highly misleading.
>
> > our comparison in this rebuttal will be restricted to LO-ARMs++ versus the ground truth dataset for the #Circles metric
>
> You should be able to find a collection of baselines trained on Guacamol in (Renz et al., 2024).
>
> > We seek further advice on these points
>
> It is not entirely clear what advice you are seeking. My advice would be to provide a best-effort comparison of your results against this (and/or other) state-of-the-art model(s) and report the corresponding results with a big asterisk listing all of the possible problems with the comparison.
>
> The main problem is that the baselines are more than five years old and even though LSTM could be an extremely hard baseline, it is hard to believe that no progress has been made since then.
>
> > the $\alpha$-$\beta$-ELBO, is a fundamentally general and essential method for improving the training stability and performance of LO-ARMs
>
> It might be that it is crucial for this method to work, and that it is widely applicable, but that does not add to its novelty.
> The change from
> $$\mathbb{E}[\log p_\theta] - D_\mathrm{KL}(q_\theta \mathbin{\|} p_\theta)$$
> to
> $$\mathbb{E}[\log p_\theta] - \beta D_\mathrm{KL}(q_\theta \mathbin{\|} p_\theta) + \alpha H[q_\theta]$$
> without a proper derivation or explanation where $\beta$ and $\alpha H[q_\theta]$ come from, does not provide much novel insights, because it almost comes down to copying the formulas from $\beta$-VAEs (Higgins, 2017) and adding some entropy regularisation (e.g. Pereyra et al., 2017), both of which are not particularly novel.
> As a matter of fact, it would probably be good to cite and point out similarities of the loss and the annealing schedules with $\beta$-VAE and LVAEs, respectively.
>
> > an application we defer to future work, as this paper focuses on improving LO-ARMs training
>
> The structure-constrained generation might be the kind of task where this architecture could make a difference.
> Maybe it would make more sense to publish the improvements proposed in this work alongside experiments in this area.
>
> > For the MOSES dataset, we directly used the raw and non-canonicalized dataset
>
> Did you do that for the baseline models as well, or only for LO-ARMS++?
> Also, did you compare performance when including canonicalisation during pre-processing?
>
>
> Currently, I am still not convinced there is enough novelty in this paper for ICLR. It is also unlikely that the promised results will change much about that. However, I would encourage the authors to tackle the task of structure-constrained generation with their model for future conferences.
>
> ### Additional References
>  - Renz et al. (2024). [Diverse hits in de novo molecule design: Diversity-based comparison of goal-directed generators.](https://pubs.acs.org/doi/full/10.1021/acs.jcim.4c00519) Journal of Chemical Information and Modeling, 64(15), 5756-5761.
>  - Higgins et al. (2017). [beta-VAE: Learning basic visual concepts with a constrained variational framework.](https://openreview.net/forum?id=Sy2fzU9gl) In International conference on learning representations.
>  - Pereyra et al. (2017). [Regularizing neural networks by penalizing confident output distributions](https://arxiv.org/abs/1701.06548). arXiv preprint arXiv:1701.06548.

---

> > ### Author Response · Authors · 2025-12-04
> > **[Follow-up discussion] Part 1/2**
> >
> > Thank you again for your swift feedback. We’re glad that you have no further concerns on, 1) enhanced performance on the GuacaMol SMILES dataset, 2)  general applicability of our method, 3) the value of learning context-dependent orderings, especially the improved orderings learned with LO-ARMs++, which would enable structure-constrained molecule generation while the standard LO-ARMs couldn’t. These are important recognition of our contributions, and thanks for your acknowledgement.
> >
> > Now we address the concerns raised in your follow-up comment. We address those on technical issues first (**Part1/2**) and then address your concerns that may be more conceptual as well as correct some factual misunderstandings an mistakes in (**Part 2/2**).
> >
> > # Other baselines on MOSES
> > Due to time constraints, we only added the standard LO-ARM without $\alpha$-$\beta$-ELBO (see Table 2 in the revised manuscript). As MOSES is a simpler dataset than GuacaMol in terms of average sequence lengths and diversity in chemical properties, we’re seeing overfitting issues with Transformer FO-ARM at the moment, but will keep finetuning FO-ARM post-rebuttal. On the other hand, **LO-ARMs++ match the performance of the top-performing models** (i.e., VAE and CharRNN/LSTM). Moreover, we see **significant performance gain with LO-ARMs++ on distributional metrics compared to standard LO-ARMs**. This is because MOSES SMILES strings are non-canonicalized, yielding higher variance due to randomized token orders, and without $\alpha$ -$\beta$-ELBO, LO-ARMs can easily collapse to premature orderings. We believe this experiment has strengthened our evaluation. In addition, we’ve also fixed the presentation issues in both Response 3 and Table 2 in the revised manuscript, and have detailed the evaluation metrics of different benchmarks in the appendix. Thank you for your comment.
> >
> > Re canonicalizing MOSES SMILES, we think this is a minor issue, as the standard dataset is uncanonicalized. Otherwise, it will be unfair to compare with other baselines in the literature.
> >
> > # LSTM as strong baseline in the literature
> > SMILES-based FO-ARMs, in particular RNN/LSTM, is widely acknowledged as the state-of-the-art model by recent work in the literature of molecule generation [e.g., 1, 2, 3], especially by the general ML community. In our work, we also aim to develop methods that are useful to the general community. Therefore, we think that LSTM serves as an appropriate baseline. Due to the same reason, we decided to defer the domain-specific application of structure-constrained generation to future work.
> >
> > Moreover, directly comparing the parameter counts of LSTM and Transformer models is inappropriate because they have fundamentally different architectures, with Transformers typically requiring more parameters than LSTMs. Our decision to fix the generator's capacity was made to ensure a fair comparison between LO-ARMs++ and the Transformer-based FO-ARMs.

---

> ### Author Response · Authors · 2025-11-26
> **Part III: Response to Q1 and Q6**
>
> First of all, we’d like to thank you for your feedback. In this reply, we will be addressing the remaining questions (i.e., Q1 and 6) raised in your initial comment, and will address your new concerns in a separate reply. Hope this makes sense to you.
>
> # Q6: How does this prefix tokenization affect other baselines?
>
> As we do not possess direct access to the LSTM implementation, we employed the Transformer-based FO-ARMs as the primary proxy architecture for ablating the effect of the prefix tokenization.
>
> **updated: corrected some errors in FO-ARM w/ prefix tokenization**
>
> FO-ARMs using vanilla/character-based and prefix tokenization show nearly identical performance on FCD and KL divergence, while the prefix tokenization downgrades validity. This is because the prefix tokenization not only increases the complexity of vocabulary, but also
> eliminates the left-to-right dependencies of matching parentheses, which yields a harder generation problem for FO-ARMs.
>
> |                                | Validity   | V.U        | V.U.N      | FCD score  | KL div     |
> |--------------------------------|------------|------------|------------|------------|------------|
> | FO-ARM w/ vanilla tokenization | 98.1 $\pm$ 0.1 | 97.8 $\pm$ 0.1 | 88.6 $\pm$ 0.3 | 87.0 $\pm$ 0.4 | 99.0 $\pm$ 0.1 |
> | FO-ARM w/ prefix tokenization  | 88.3 $\pm$ 0.7 | 83.1 $\pm$ 0.2 | 82.8 $\pm$ 0.3 | 87.2 $\pm$ 0.2 | 99.1 $\pm$ 0.1 |
>
> Moreover, from the another perspective, the character-based tokenization implicitly assumes a left-to-right token order through parsing individual parentheses as tokens, and therefore, it is
> compatible with fixed left-to-right ARMs. On the other hand, the prefix tokenization may not be compatible with FO-ARMs, as it doesn't assume a fixed left-to-right orders.
>
> A more detailed ablation analysis can be found in Section 3.1 and Appendix D.3 in the revised manuscript.
>
> # Q1: Evaluation on the #Circles metric
>
> We appreciate the suggestion to incorporate this additional metric. We are pleased to report that we've fixed the code, and we now present the preliminary evaluation results for the #Circles metric on the GuacaMol SMILES dataset. Given that this metric has not yet achieved widespread adoption in the literature, our comparison in this rebuttal is primarily confined to FO-ARMs, LO-ARMs++, and the ground truth GuacaMol dataset, necessitated by time constraints. The complete table will be furnished in the final revised manuscript.
>
> Specifically, in our evaluation, we set the threshold t=0.75 as suggested by [1], each evaluated data point was evaluated with 1000 SMILES strings, and for each model we collected 1000 data points.
>
> Furthermore, in response to the paper Renz et al. (2024) suggested in your recent comment, we note that their evaluation focused on goal-oriented generation tasks, which precludes a direct comparison with our results from unconditional molecule generation.
>
>
> |                          | Dataset          | FO-ARMs w/ char-based tokenization | FO-ARMs w/ prefix tokenization | LO-ARMs++         |
> |--------------------------|------------------|------------------------------------|--------------------------------|-------------------|
> | #Circles (threshold=0.75) | 46.1 $\pm$ 12.6% | 42.2 $\pm$ 12.4%        | 42.1 $\pm$ 11.9%             | 45.6 $\pm$ 12.2% |
>
> As is evident from the table, the FO-ARMs utilizing either character-based or prefix tokenization exhibit nearly identical performance with respect to the #Circles metric, thereby aligning with our response to Q6. A distinct performance improvement is observed for LO-ARMs++ when compared against FO-ARMs, with its result demonstrating only a marginal difference from the ground truth dataset.
>
> [1] https://openreview.net/pdf?id=Yo06F8kfMa1

---

> ### Author Response · Authors · 2025-12-04
> **[Follow-up discussion] Part 2/2**
>
> # Unsuitable direct comparison with your recommended papers
> As chemistry is not our expertise neither, and you seemed fairly confident in both criticizing our evaluation and recommending relevant papers, we highly valued your opinions, read the papers carefully and implemented the #Circles metric suggested by you. However, for the two papers you recommended (i.e., Özçelik et al. (2024) and Renz et al. (2024)), we find that they are methodologically unsuitable for direct comparison due to using different dataset and evaluation sample size, and more importantly on different tasks (Please see our previous responses for more detail). Therefore, we asked for your advice and had wanted to get involved in a more constructive discussion before implementing your comments. If you don’t claim your expertise in the chemistry domain, we’d be more than happy to have a more open discussion with you on these issues.
>
>
> # Re lack of proper derivation and explanation of developing $\alpha$-$\beta$-ELBO
> We respectfully disagree on this point. We clearly justified our motivation and derivation in the Method section (Section 2), which is also one of the strengths of our work recognized by other reviewers. Moreover, we clearly cited Higgins et al. (2017) (Line#256) and discussed its connection with our method. For Pereyra et al. (2017), we are aware of this paper, however, we didn’t cite in the initial version, as we address different problems, i.e., supervised learning and unsupervised learning.
>
> # Re your concern about novelty
> We appreciate the recommendation of Higgins et al. (2017) and Pereyra et al. (2017), both of which, in your opinion, may “not be particularly novel”. Regarding their novelty, we consider this to be subjective, and therefore take no personal position on these papers. However, we must note that the value and contributions of these two papers to the general ML community have been highly recognized, not only through the fact that they were accepted by ICLR but also through their high citation counts. The ML community tends to favor straightforward and efficient methods over intricate ones, as simple yet effective approaches can be rapidly adopted and easily generalized across various research domains. Therefore, we believe our paper aligns well with the criteria for the ICLR conference in terms of our results and contributions.
>
>
> # Partially cited response
> In our responses, we are very consistent in and specific about that we only enhanced our results on the GuacaMol benchmark, and that the results on the MOSES are just preliminary. Therefore, we purposefully split the results into two responses. We also highlighted the performance difference on different distributional metrics on MOSES. However, you only cited **part of the title of Response 2**, which is misleading.
>
> [1] https://arxiv.org/abs/2209.14734
>
> [2] https://arxiv.org/pdf/2410.04263
>
> [3] Renz et al. (2024). Diverse hits in de novo molecule design: Diversity-based comparison of goal-directed generators. Journal of Chemical Information and Modeling, 64(15), 5756-5761.
>
> [4] Özçelik et al. (2024). Chemical language modeling with structured state space sequence models. Nature Communications, 15(1), 6176.

---

### Official Review · Reviewer_iSaf · 2025-11-03

**Soundness:** 2
**Presentation:** 2
**Contribution:** 2
**Rating:** 4
**Confidence:** 3

**Summary:**

The paper proposes improvements to learned-order autoregressive models which explore more generation orders and stabilize training, yielding better test-set log-likelihood and better distribution learning. They derive a tighter ELBO that unifies fixed-order and any-order autoregression, and add an entropy regularization term, with new terms weighted by hyperparameters $\alpha$ and $\beta$. Annealing large $\alpha$ and $\beta$ during training allows the model to initially explore different orders until eventually exploiting to optimize ELBO. The paper verifies that these changes do result in a less-greedy order policy. The authors also find that applying dropout to attention scores disrupts training, and instead apply dropout to the output of the attention layer. In experiments on distribution learning GuacaMol, LO-ARM++ outperforms LO-ARM in FCD and NLL, and is competitive with FO-ARMs. LO-ARM++ also finds interpretable generation orders.

**Strengths:**

The paper well-characterizes the learned generation order with informative figures.

Extensive ablations show how the proposed changes affect the resulting model's entropy and training stability.

The paper shows how its contributions generalize other works such as FO-ARMs and AO-ARMs.

**Weaknesses:**

Multiple tricks are employed to improve performance specifically on generating SMILES, which casts doubt on the claim of generalizable improvements over LO-ARMs. A new tokenization is required for LO-ARM to compete with fixed-order autoregressive models, as well as filtering out uncommon tokens. The experimental setup is also restricted to just canonical SMILES, but it is known that random SMILES orders boosts performance.

A pessimistic criticism is that the new $\alpha$-$\beta$-ELBO simply provides extra hyperparameters for achieving better performance than previous models.

**Questions:**

1. In Table 1, why does Transformer FO-ARM underperform LSTM ARM?
2. Why is test NLL reported as an upper bound, when FO-ARM provides exact NLL?
3. Why is FO-ARM NLL on padded SMILES not comparable to LO-ARM NLL? Can't you exclude the contribution of log-likelihood of pad tokens?
4. Is the annealing schedule expected to always provide a benefit, or is it dependent on the domain?
5. To support the claim that LO-ARMs are generally improved by the new regularization terms, there should be at least one other set of experiments on a different domain such as text or Sudoku, or even toy data on traversing graphs.
6. A body of work studying token-ordering in autoregressive / discrete diffusion models is not cited. These works mention experiments such as Sudoku which may be able to better demonstrate the utility of learned-order autoregression.

[1] Kim, J., Shah, K., Kontonis, V., Kakade, S., & Chen, S. (2025). Train for the worst, plan for the best: Understanding token ordering in masked diffusions. arXiv preprint arXiv:2502.06768.

[2] Bachmann, G., & Nagarajan, V. (2024). The pitfalls of next-token prediction. arXiv preprint arXiv:2403.06963.

nit-picking
1. There is a missing comma in the $D_{KL}$ in Equation 3.
2. Table 2: "SMILES strsings"
3. line 640: "LO-AMRs+"
4. line 647: "highlighy"
5. line 809: repeated sentence

---

> ### Author Response · Authors · 2025-11-24
> **Part I**
>
> We sincerely thank you for your positive feedback, and we are addressing your comments below. Moreover, we rearrange the order of our responses to the corresponding weaknesses and questions to make our presentation more coherent.
>
> **[W: weakness point, Q: question]**
>
> # W2: lack of evaluation on non-canonical SMILES data
> First, we improved results on the GuacaMol benchmark, achieving SOTA on distribution learning (see Response 2). We note that canonicalization is standard for the baselines in the literature. To evaluate LO-ARMs++ on non-canonical and random SMILES, we ran experiments on the MOSES dataset. Our preliminary results showed that in terms of distribution learning, except for FCD, LO-ARMs++ outperform CharRNN on all other distribution learning metrics (see Response 3).
>
> # Q5: re the general effectiveness of $\alpha$-$\beta$-ELBO
>
> To address this specific comment, in addition to the evaluation on MOSES, we incorporated supplementary evaluations on two distinct molecular graph generation tasks, namely the ZINC250k and GuacaMol graphs, as detailed in Response 4. These experiments on molecular graph generation were strategically designed to isolate the effect of the $\alpha$-$\beta$-ELBO technique by directly utilizing the standard tokenization and GraphTransformer architecture provided within the literature. Across both benchmarks, the inclusion of the $\alpha$-$\beta$-ELBO demonstrably yielded positive improvements in both validity and FCD.
>
> Given these extensive evaluations, which span both string-based (SMILES) and graph-based molecular generation modalities, we confidently conclude that LO-ARMs are generally and effectively improved through the application of $\alpha$-$\beta$-ELBO.
>
> # W3: re the criticism about$\\alpha$-$\\beta$-ELBO introducing extra hyperparameters to achieve better performance
>
> We respectfully disagree with this criticism. The core advantage of the $\alpha$-$\beta$-ELBO is its simplicity of enabling the implementation of curriculum learning without altering the final learning objective. This technique is conceptually akin to established methods, such as the warm-up period for the KL-divergence term in VAEs [1], yet it directly and effectively addresses the instability issues inherent in training LO-ARMs. Furthermore, our approach is demonstrably effective and scalable, as evidenced by its strong performance across both SMILES and molecular graph generation tasks in our extended evaluation.
>
> [1] Sønderby, C. K., Raiko, T., Maaløe, L., Sønderby, S. K., & Winther, O. (2016). Ladder variational autoencoders. Advances in neural information processing systems, 29.
>
> # Q4: Is the annealing schedule expected to always provide a benefit?
>
> The annealing schedule solves two key training problems: 1) instability from the q distribution's premature convergence, and 2) convergence to severe local optima when the variational q policy is too greedy. This domain-agnostic schedule is expected to be more helpful for larger domains (longer sequences, bigger vocabulary) where these issues are more acute.
>
> We list the maximum sequence length and the number of node types for all datasets below
>
> |                 | max sequence length | vocabulary/node types |
> |-----------------|---------------------|-----------------------|
> | GuacaMol SMILES | 96                  | 129                   |
> | MOSES SMILES    | 47                  | 61                    |
> | GuacaMol Graphs | 3096                | 12                    |
> | ZINC250k Graphs | 1482                | 9                     |
>
>
> As we can see in the case of molecular graph generation, on smaller or simpler domains (i.e.., ZINC250k), the gain of LO-ARMs++ or $\\alpha$-$\\beta$-ELBO may be marginal, while on larger domain its gain is more distinctive, especially GuacaMol Graphs vs ZINC250k Graphs.
>
> # Q1: Re the performance difference between FO-ARM and LSTM ARM
>
> We acknowledge the performance gap. Our FO-ARM uses a GPT-2 Transformer with Relative Positional Encoding (RoPE), and its performance aligns with GPT-RoPE results in [1]. Note that it is not uncommon that recurrent neural networks outperform Transformers on non-language modeling tasks, as shown in Table 4 of [2].
>
> Our research did not focus on improving the Transformer for SMILES generation. We maintained the same Transformer architecture across our baselines (AO-ARMs, FO-ARMs) for consistency, with LSTM remaining our primary competitor. We leave the work of adapting the Transformer to SMILES generation to future work, which would further improve the performance of LO-ARMs++.
>
> [1] https://www.nature.com/articles/s41598-025-86840-z/tables/1
>
> [2] Gu, A., Goel, K., & Ré, C. (2021). Efficiently modeling long sequences with structured state spaces. arXiv preprint arXiv:2111.00396.

---

> ### Author Response · Authors · 2025-11-24
> **Part II**
>
> # Q2: re the inappropriate NLL reports for FO-ARMs using lower bound.
>
> Thanks a lot for your thorough review. This is indeed a mistake, and we’ll fix it in the table in the updated PDF.
>
> # Q3: re the difference in NLL between FO-ARM and LO-ARMs.
>
> Thank you for your comment. We’re rerunning experiments to exclude the effect of the paddings in FO-ARMs, which will be updated in the revised PDF.
>
> # Q6: missing references and typos.
>
> Thanks very much again for your thorough review and advice, and we’ll update the PDF accordingly.

---

> ### Author Response · Authors · 2025-11-26
> **Part III: Q3 Cont'd**
>
> # Q3: Re the difference in NLLs between FO-ARM and LO-ARMs (Cont'd)
>
> Thank you again for noting this detail. This incomparability is casued by different length-conditioning setups of FO- and LO-ARMs.
>
> We have re-run experiments and reconsidered excluding padding tokens during NLL calculation. In Fixed-Order (FO) left-to-right ARMs, padding tokens act like End-of-Sequence (EOS) tokens; once predicted, subsequent padding is almost certain. To show this, we compared the NLLs on the test set for full padded sequences and non-padding tokens after training converged.
>
> The table below shows that NLL values for the full masked sequence and non-padding tokens are nearly identical for both FO-ARMs using character-based and prefix tokenization, confirming that both models nearly perfectly predict padding tokens.
>
> |                                    | NLL on full padded sequence | NLL on non-padding tokens |
> |------------------------------------|-----------------------------|---------------------------|
> | FO-ARMs w/ char-based tokenization | 32.78                       | 32.42                     |
> | FO-ARMs w/ prefix-tokenization     | 35.04                       | 34.94                     |
>
> Explicitly adding an EOS token for SMILES to fully eliminate padding effects on NLL would unnecessarily increase the vocabulary and sequence length.  Thus, comparing AO-ARMs and LO-ARMs using NLLs is more appropriate, as they use the same length conditioning. To simplify and make data preprocessing consistent for our own baselines, for LO-ARMs vs. FO-ARMs, we prioritize the metrics that evaluate the final generated SMILES strings. We hope this makes sense to you.
>
> We will update the PDF accordingly.
>
> We believe we have fully addressed your concerns, justifying a more favorable assessment. Please don’t hesitate to let us know if you have any questions.

---

> ### Author Response · Authors · 2025-12-04
> **Thank you**
>
> Dear Reviewer,
>
> We've now updated our manuscript to resolve the following remaining issues in your feedback:
>
> 1) Typos.
> 2) Missing references.
> 3) Refined presentation in the tables, especially adding stddev for all our models. Note that, the stddev of the baseline results in the literature are still missing, as they didn't report. Moreover, to avoid confusion, we removed the comparision of NLLs as they are not directly comparable due to different length conditioning and tokenizations, especially between FO-ARMs and LO-ARMs.
>
> We believe that we've fully resolved your comments. Thank you so much for your constructive feedback.
>
> Best,
>
> Authors

---

### Author Response · Authors · 2025-11-24
**Shared Response 1/4: Clarification on Motivation and general usefulness of our method**

We would like to thank all reviewers for their detailed and insightful comments. In these rebuttals,, we have included additional experiments and updated results. Below we respond to the shared comments that were raised by multiple reviewers.

First, we have improved the performance of LO-ARMs++, achieving SOTA on the GuacaMol SMILES dataset (Rebuttal table 1 in Shared Response 2), surpassing LSTMs. We believe LO-ARMs++ is generally useful for new generative molecule models and will open source the code upon acceptance.

Second, in addition to the GuacaMol SMILES dataset, we’ve enhanced the evaluation with extra benchmarks. Here is the summary of the datasets. Results are shown in the remaining shared responses.

| Modality | Dataset  | Notes                                                              |
|----------|----------|--------------------------------------------------------------------|
| SMILES   | Gucamol  | canonicalized                                                      |
|          | MOSES    | non-canonicalized                                                  |
| Graph    | ZINC250k | #samples: 250k, #nodes: [6, 38], #node types: 9, input dims: 1482  |
|          | GuacaMol | #samples: 1.2M, #nodes: [2, 62], #node types: 12, input dims: 3906 |

Finally, the main contribution is improving and stabilizing LO-ARMs training via $\alpha$-$\beta$-ELBO, enabling curriculum learning without changing the training objective. This improvement is critical for performance with larger datasets, as shown later.

Note that, in the GuacaMol SMILES benchmark table, the FCD is normalized as a standard in the literature, and all other reports of FCDs are unnormalized. Moreover, as Reviewer cn57 suggested, the novelty measure may not be very meaningful, and therefore, we prioritize the comparison against the distribution learning metrics and keep novelty as a reference as it’s a standard metric reported in the literature.

---

### Author Response · Authors · 2025-11-24
**Shared Response 2/4: New SOTA performance of LO-ARMs++ on the GuacaMol benchmark and refined presentation of Table 1**

We enhanced LO-ARMs++ performance, surpassing LSTM and achieving state-of-the-art results, by increasing the variational q network from 1 to 6 layers for better posterior estimation. The capacity of the generative model (p network) remains the same, ensuring unchanged inference cost.

**Please see Table 1 & 4 in the manuscript for full comparision**

*Rebuttal table 1: Improved results on the GuacaMol SMILES dataset*
| Method         | Modality | Validity | Uniqueness | Novelty | FCD score $\uparrow$  | KL div $\uparrow$ |
|----------------|----------|----------|------------|---------|------|--------|
| Random sampler |          | 100.0    | 99.7       | 0       | 92.9 | 99.9   |
| DiGress [3]    | Graph    | 85.2     | 85.2       | 85.1    | 68.0 | 92.9   |
| DeFog [2]      | Graph    | 99.0     | 99.0       | 97.9    | 73.8 | 97.7   |
| LSTM [1]       | SMILES   | 95.9     | 95.9       | 87.5    | 91.3 | 99.1   |
| LO-ARMs++      | SMILES   |     93.9 $\pm$ 0.2     |     93.9 $\pm$ 0.2       |   85.9 $\pm$ 0.3      |  **91.4 $\pm$ 0.1**    |   **99.2 $\pm$ 0.1**     |

Our previous Table 1 followed the standard literature reporting style (e.g., [1, 2, 3]), showing only one number, including the GuacaMol benchmark's LSTM baseline. Since the literature uses 10,000 ground truth samples for FCD evaluation, we must use the same number for comparable results. (the FCD has a sample size-dependent bias).

We generated 5 batches of ~15,000 valid LO-ARMs++ samples. For each batch, we ran five evaluations by randomly sampling 10,000 generated and 10,000 ground truth samples from the test set. We report the means and standard deviations across these 5 batch evaluations.

In the revised table, we also report KL divergence, which measures the difference in probability distributions for various physicochemical descriptors. Small KL divergence values indicate the model successfully captures the training set's molecular distributions [1].

The GuacaMol benchmarks use the training set as the reference for distribution learning metrics. Since baselines (like LSTM) memorize training data (shown by their novelty), this can cause bias from overfitting. To address this, our work uses the test set as the reference data for evaluation.

References
[1] https://arxiv.org/abs/1811.09621

[2] https://arxiv.org/pdf/2410.04263

[3] https://arxiv.org/pdf/2209.14734

[4] https://github.com/BenevolentAI/guacamol/blob/master/guacamol/data/get_data.py#L156

[5] https://github.com/molecularsets/moses

[6] https://arxiv.org/abs/2503.05979

---

### Author Response · Authors · 2025-11-24
**Shared Response 3/4: Extended evaluation on the MOSES benchmark**

The raw standard GuacaMol dataset is already canonicalized [4]. To show LO-ARMs++'s effectiveness, we provide preliminary results on the MOSES benchmark [5], which uses non-canonicalized SMILES data.

We used the MOSES evaluation library and reported standard distribution learning metrics against the test set. Unlike GuacaMol's FCD, MOSES's FCD is unnormalized (smaller is better). Metrics like lipophilicity (logP), Synthetic Accessibility (SA), and QED measure the Wasserstein-1 distance between generated and test set molecule distributions.

*Rebuttal table2: Results on the MOSES SMILES dataset* **Updated: highlight all best resultsm in this response, full comparision can be found in Table 2 in the revised manuscript.**
| Method    | FCD  $\downarrow$           | SNN   $\uparrow$            | Frag $\uparrow$          | Scaf    $\uparrow$         | IntDiv    $\uparrow$       | IntDiv2  $\uparrow$      | logP $\downarrow$         | SA  $\downarrow$          | QED   $\downarrow$          | Filters   $\uparrow$       | Validity  $\uparrow$       | Novelty    $\uparrow$      |
|-----------|-----------------|-----------------|---------------|-----------------|-----------------|---------------|---------------|---------------|-----------------|-----------------|-----------------|-----------------|
| CharRNN   | **0.0732$\pm$0.0247**   | 0.6015$\pm$0.0206   | 0.9998$\pm$0.0002 | 0.9242$\pm$0.0058	   | 0.8562$\pm$0.0005   | 0.8503$\pm$0.0005	 |         0.057 |         0.016 |          0.0022 | 0.9943$\pm$0.0034   | **0.9748$\pm$0.0264**	   | **0.8419$\pm$0.0509**   |
| LO-ARMs++ | 0.1390 $\pm$ 0.0043 | **0.6119 $\pm$ 0.0001** | **1.0 $\pm$ 0**       | **0.9286$\pm$0.0027** | **0.8586 $\pm$ 0.0020** | **0.8543$\pm$ 0.0013**   | **0.047 $\pm$ 0.002** | **0.012 $\pm$ 0.001** | **0.0015 $\pm$ 0.0003** | **0.9951 $\pm$ 0.0002** | 0.9459 $\pm$ 0.0137 | 0.8012 $\pm$ 0.0213 |

As we can see in the table, in terms of distribution learning, except for FCD, LO-ARMs++ outperform CharRNN on all other distribution learning metrics.

References
[1] https://arxiv.org/abs/1811.09621

[2] https://arxiv.org/pdf/2410.04263

[3] https://arxiv.org/pdf/2209.14734

[4] https://github.com/BenevolentAI/guacamol/blob/master/guacamol/data/get_data.py#L156

[5] https://github.com/molecularsets/moses

[6] https://arxiv.org/abs/2503.05979

---

### Author Response · Authors · 2025-11-24
**Shared Response 4/4: Supplementary evaluation on molecular graph generation**

Our main contribution is improving and stabilizing LO-ARMs training via $\alpha$-$\beta$-ELBO. This technique is modality-agnostic, and we show its impact on molecular graph generation benchmarks (ZINC250k and GuacaMol) by reusing standard tokenization and GraphTransformer from the literature. For a fair comparison on ZINC250k, our LO-ARMs++ reimplementation sets alpha=1 and beta=0 to reproduce LO-ARMs.

Preliminary results show $\\alpha$-$\\beta$-ELBO improves validity and FCD on both benchmarks. The gain is more significant on GuacaMol, which is a more challenging dataset (max node number = 68) compared to ZINC250k (max node num=32). This suggests that curriculum learning with $\alpha$-$\beta$-ELBO is crucial for encouraging exploration and finding better local optima, especially in larger domains like GuacaMol graphs.

Note that we are yet to incorporate any graph-specific improvements (e.g., alternative Graph Transformer architectures [2]), which  are complementary to the $\\alpha$-$\\beta$-ELBO.

*Rebuttal table3: Preliminary results on the molecular graph benchmarks*
| Dataset  | Model                      | Validity     | Novelty     | Uniqueness  | FCD $\downarrow$         |
|----------|----------------------------|--------------|-------------|-------------|-------------|
| ZINC250k | LO-ARMs [6]                | 95.97 $\pm$ 0.27 | 100.0 $\pm$ 0.0 | 100.0 $\pm$ 0.0 | 3.33 $\pm$ 0.13 |
|          | LO-ARMs (reproduced)      | 95.14 $\pm$ 0.71 | 100.0 $\pm$ 0.0 | 100.0 $\pm$ 0.0  | 3.49 $\pm$ 0.10 |
|          | LO-ARMs w/ $\\alpha$-$\\beta$-ELBO | **96.94** $\pm$ 0.33 | 100.0 $\pm$ 0.0 | 100.0 $\pm$ 0.0 | **3.27 $\pm$ 0.04** |
| GuacaMol | LO-ARMs                    | 92.36 $\pm$ 0.59 | 100.0 $\pm$ 0.0 | 100.0 $\pm$ 0.0  | 5.21 $\pm$ 0.06 |
|          | LO-ARMs w/$\\alpha$-$\\beta$-ELBO  | **95.31** $\pm$ 0.56 | 100.0 $\pm$ 0.0 | 100.0 $\pm$ 0.0  | **3.97 $\pm$ 0.03** |

References
[1] https://arxiv.org/abs/1811.09621

[2] https://arxiv.org/pdf/2410.04263

[3] https://arxiv.org/pdf/2209.14734

[4] https://github.com/BenevolentAI/guacamol/blob/master/guacamol/data/get_data.py#L156

[5] https://github.com/molecularsets/moses

[6] https://arxiv.org/abs/2503.05979

---

### Author Response · Authors · 2025-12-04
**Summary of Rebuttals**

Dear AC,

Thank you for managing the review process and to all reviewers for their insightful feedback. We believe our manuscript has significantly improved through addressing their concerns. For your convenience, we summarize our core contribution and how we addressed reviewer comments. Please see our responses in the corresonding threads.

**Core contributions**

* Conceptually, we introduce the Generalized Next-Token-Predictor (NTP), connecting fixed-order (FO), any-order (AO) and learning-order (LO) autoregressive models (ARMs) under a unified variational inference framework.

* Technically, stemming from the generalized NTP, we introduce $\alpha$-$\beta$-ELBO, an improved training loss, which stabilizes training LO-ARMs and allows for implementing an exploration-exploitation strategy for unsupervised learning with our altering learning objective. This forms the basis of the LO-ARMs++ model training procedure.

* The general applicability of the $\alpha$-$\beta$-ELBO is demonstrated by its critical role in enhancing generation performance, particularly in distributional metrics, across various molecular generation benchmarks. This includes both molecular graph (ZINC250k and GuacaMol graphs) and sequence-based (GuacaMol and MOSES SMILES strings) generation tasks.

* In practical terms, we introduce two reusable engineering enhancements that complement the $\alpha$-$\beta$-ELBO. These improvements achieve **state-of-the-art results** on unconditinal generation task in the GuacaMol SMILES benchmark, outperforming existing fixed-order (left-to-right) autoregressive models based on both transformer and LSTM architectures. Furthermore, we achieved performance parity with RNN models on the MOSES benchmark.

* LO-ARMs++ not only achieves state-of-the-art generation performance based on distribution metrics but also **learns coherent and informative** generation orderings. These sequences effectively **uncover molecular structures** without relying on pre-existing inductive biases. Furthermore, the enhanced ordering learned by LO-ARMs++ (compared to standard LO-ARMs) **facilitates self-refinement** during the molecule generation process (see Fig 1 in the manuscript).

During rebuttal, we add extensive new experiments and discussions to adequately address the reviewers' questions one by one. All these new results are included into the revision.

**Enhanced experiments and improved presentation**
* [R#BXeH, R#cn57] Enhanced performance on the unconditional generation task in the GuacaMol SMILES benchmark, yielding **state-of-the-art performance**, surpassing the performance of both Transformer- and LSTM-based FO-ARMs.

* [R#iSaf, R#WFFz, R#BXeH, R#cn57] Enriched evaluation via adding initial MOSES SMILES benchmark results. LO-ARMs++ outperforms standard LO-ARM and matches RNN performance, even with high variance from randomized SMILES strings, further confirming the effectiveness of $\alpha$-$\beta$-ELBO.

* [R#iSaf, R#WFFz, R#BXeH, R#cn57] Supplementary experiments on GuacaMol and ZINC250k graph generation show $\alpha$-$\beta$-ELBO improves LO-ARMs, confirming its **general applicability and practical effectiveness** across SMILES and graph-based generation.

* [R#iSaf, R#WFFz] Unlike the standard reports in the literature, for all our results, we report both mean and standard deviation in all our relevant tables.

**Extended analysis**:
* [R#cn57] Ablation analysis on prefix tokenization for FO-ARMs and LO-ARMs.

* [R#BXeH] Sensitivity analysis on the hyperparameters for annealing schedules.

* [R#WFFz] Effect of length prior on performance.

* [R#cn57] Evaluation with the #Circles metric. We provide additional evaluation on the #Circles metric, the results of which align with those on other distributional metrics.

**Conceptual questions**:
* [R#iSaf] Is the annealing schedule expected to always provide a benefit? Through our extended evaluation, we can see that $\alpha$-$\beta$-ELBO is generally beneficial, and can yield larger performance gains in more complex domains, avoiding training to collapse to premature local optima.

* [R#cn57] Value of learned orderings and possible downstream tasks.

* [R#WFFz] Could an asynchronous schedule for $\alpha$ and $\beta$ be even better?

* Other minor questions are either due to reviewers' misunderstanding or could be easily addressed as done in the revision.

In addition, we’ve also pointed out some critical factual misunderstandings and mistakes in the comments raised by Reviewer cn57.

All these changes have been incorporated into the revised manuscript, highlighted in red for your convenience.

To summarize, we have addressed every single concern raised by reviewers in the rebuttal revision. We believe the quality of the revised manuscript has been greatly improved, with enhanced SOTA results and analysis, and our contributions are well suited in the context of ICLR.

Best,

Authors

---

### Meta-Review · Area_Chair_F2B4 · 2026-01-02

**Summary:**

This work introduces a new method, LO-ARMs++, for molecular string generation. The targeted problem is fixed-order autoregressive models, which may not be the correct answer for molecular generation. The authors therefore proposes a way over the learning-order ARMs and introduce LO-ARMS++, an enhanced version and the improved training method \alpha-\beta-ELBO contributes the most. Experiments compared to some baselines show the effectiveness of the new training method.

Reviewers raised multiple concerns and questions about this work, from the method to the experiments. Some concerns about the compared baselines, experiments, and the conceptual notes are partially solved. The rebuttal makes efforts and reviewers give feedback in some way. However, there are still lots of concerns remained, such as the novelty and the design rationality. Besides, the experiments also raise quite concerns though the "SOTA" achieved.

**Reviewer Concerns:**

Some concerns about the compared baselines, experiments, and the conceptual notes are partially solved. The rebuttal makes efforts and reviewers give feedback in some way. However, there are still lots of concerns remained, such as the novelty and the design rationality. Besides, the experiments also raise quite concerns though the "SOTA" achieved.

**Reviewer Scores:**

According to the current version, Reviewer iSaf is possible to increase the score. However, Reviewer cn57 and BXeH are hard to improve their score to a positive one, though some misunderstandings are resolved. The concerns remain heavy from their further comments during rebuttal.

---

### Decision · Program_Chairs · 2026-01-26

Reject